# Discovery and pharmacophoric characterization of chemokine network inhibitors using phage-display, saturation mutagenesis and computational modelling

Serena Vales [1,3], Jhanna Kryukova [1,3], Soumyanetra Chandra [1], Gintare Smagurauskaite[1], Megan Payne [1], Charlie J. Clark [1], Katrin Hafner[1], Philomena Mburu[1], Stepan Denisov [1], Graham Davies[1], Carlos Outeiral [2], Charlotte M. Deane [2], Garrett M. Morris [2] & Shoumo Bhattacharya [1] ✉

CC and CXC-chemokines are the primary drivers of chemotaxis in inflammation, but chemokine network redundancy thwarts pharmacological intervention. Tick evasins promiscuously bind CC and CXC-chemokines, overcoming redundancy. Here we show that short peptides that promiscuously bind both chemokine classes can be identified from evasins by phage-display screening performed with multiple chemokines in parallel. We identify two conserved motifs within these peptides and show using saturation-mutagenesis phage-display and chemotaxis studies of an exemplar peptide that an anionic patch in the first motif and hydrophobic, aromatic and cysteine residues in the second are functionally necessary. AlphaFold2-Multimer modelling suggests that the peptide occludes distinct receptor-binding regions in CC and in CXC-chemokines, with the first and second motifs contributing ionic and hydrophobic interactions respectively. Our results indicate that peptides with broad-spectrum anti-chemokine activity and therapeutic potential may be identified from evasins, and the pharmacophore characterised by phage display, saturation mutagenesis and computational modelling.

Chemokines are structurally related secreted proteins that drive the migration of neutrophils, monocytes, T-cells and other leucocytes during the inflammatory response to injury or infection and in immune system homoeostasis[1]. They play a major role in immune-inflammatory diseases ranging from atherosclerosis, rheumatoid arthritis, inflammatory bowel disease and cytokine-storm[2-5]. The 46 human chemokines are grouped into CC, CXC, CX3C and XC classes based on the spacing of their N-terminal cysteine residues[1]. CC-chemokines constitute the largest class, with 26 members in humans, while CXC chemokines have 17 members[1]. The binding of chemokines to a family of 18 G-protein coupled receptors (GPCRs) activates signalling, leading to chemotaxis or directed migration of leucocytes to the site of chemokine expression[1]. Chemokines are also functionally grouped into inflammatory or homoeostatic types with inflammatory chemokines causing leucocyte recruitment to the sites of inflammation, and constitutively expressed homoeostatic chemokines mediating immune cell homoeostasis[1]. Several homoeostatic chemokines also have inflammatory roles, and thus have dual functions[1,6].

Chemokines have a conserved disulfide-bonded tertiary structure with an unstructured N-terminal domain, followed by a N loop, a

[1]Wellcome Centre for Human Genetics and RDM Cardiovascular Medicine, University of Oxford, Roosevelt Drive, Oxford OX3 7BN, UK. [2]Department of Statistics, University of Oxford, 24-29 St Giles, Oxford OX1 3LB, UK. [3]These authors contributed equally: Serena Vales, Jhanna Kryukova. ✉e-mail: shoumo.bhattacharya@cardiov.ox.ac.uk

three-stranded β-sheet (linked by loops, known as 30s−between β1 and β2, and 40s−between β2 and β3), and an α-helix[7]. A consensus view is that chemokine receptors engage with chemokines through four chemokine recognition sites (CRS) – 0.5, 1, 1.5 and 2[7]. CRS0.5, i.e., the receptor distal N-terminus, interacts with the β1-strand. CRS1, located in the receptor N-terminus, interacts with the N and 40 s loops of the chemokine. CRS1.5, in the receptor N-terminus between CRS1 and CRS2, targets the conserved disulfide of the chemokine[7]. CRS2, i.e., the transmembrane pocket of the receptor, interacts with the N-terminus of the chemokine[7]. Chemokines often homodimerize, with CC-chemokines dimerising by interactions between CC motifs and preceding N-terminal residues, while CXC-chemokines dimerise by interactions between the β1-strands[7]. Chemokines also hetero-dimerize, with CC-type heterodimers activating and CXC-type heterodimers inhibiting chemokine activity[8]. Dimerisation impacts receptor binding: CC-class chemokines cannot bind their receptors as dimers, whereas CXC-class chemokines can[7].

The chemokine network is characterised by redundancy and robustness to genetic or environmental variation[9]. Functional redundancy is a driver of biological network robustness[10], and inflamed tissues are characterised by the expression of multiple chemokines with functionally redundant activities[11]. For instance, in rheumatoid arthritis, 15 CC and 11 CXC chemokines are expressed in joint tissues[3]. The human atherosclerotic plaque expresses 16 CC and 7 CXC chemokines[11]. Functional redundancy in chemokine-driven inflammatory leucocyte recruitment is likely created by promiscuous (i.e., "one-to-many") interactions between chemokines and their receptors[12], and the expression of multiple chemokine receptors by each leucocyte class[13]. The robustness to environmental variation has frustrated the development of therapeutics that target the chemokine network in inflammatory disease[3,14].

Following initial observations on the modulation of host immunity by ticks (reviewed in ref. 15), chemokine-binding proteins were identified in the saliva of Ixodid ticks[16]. Molecular cloning identified three chemokine-binding proteins, referred to as evasins, from the salivary glands of the dog tick *Rhipicephalus pulchellus*[17,18]. Evasins EVA1 and EVA4 specifically bind CC chemokines, whereas EVA3 specifically binds CXC chemokines[17,18]. Fifty evasin proteins from diverse tick species have been expressed and characterised to date[19–24]. They may be classified into two functional groups, A and B, with exclusive binding to either CC- or CXC- chemokines, respectively[25]. Sequence-based phylogenetic analysis of tick transcriptomes indicates that CC-chemokine binding class A evasin-like proteins segregate into three subclasses, A1, A2 and A3[25,26]. Structural characterisation shows that class A1 evasins have a four-disulfide-bonded core structure, with CC-chemokine binding specificity arising within the CC motif itself[27–29], whereas A3 evasins have a five-disulfide-bonded core structure[26]. Class B evasins have a three-disulfide-bonded "knottin" structure[23].

An important feature of both class A and class B evasins is that they typically bind and inhibit *multiple* chemokines in a "one-to-many" or promiscuous manner[19–23,25]. This facility, achieved by natural selection in ticks over several million years, very likely underlies their ability to maintain the prolonged skin attachment needed for blood feeding, despite local chemokine expression[25]. A key challenge is to exploit or mimic the ability of evasins to bind multiple chemokines for developing therapeutics that target the chemokine network in disease. Studies with evasins have shown that they are effective in diverse inflammatory disease models including arthritis, myocardial infarction, lung inflammation, pancreatitis and psoriasis (reviewed in ref. 30). Despite success in models of inflammatory disease, evasins have not been clinically developed as therapeutics, at least in part due to the perceived obstacles of foreign proteins as biological therapeutics such as immunogenicity, coupled with relatively high manufacturing costs, and binding specificity requiring that both class A and B evasins be used in combination. Alternative approaches to target

chemokines have exploited evolution of broadly cross reactive antibodies using yeast surface display[31], and the isolation of promiscuous chemokine-binding peptides using phage-display followed by conversion into peptibody (IgG1-Fc) fusions[32].

Peptides and peptidomimetics that mimic the activity of parental proteins have been successfully developed from several naturally occurring proteins into viable clinical therapeutics. Exemplars include the peptidomimetics captopril, eptifibatide and tirofiban, which are based on peptides derived from the snake venom toxins bradykinin-potentiating-peptide, barbourin and echistatin respectively[33]. Peptides derived from the chemokine CCL5 have been shown to bind and inhibit chemokine function in vitro and in vivo by interfering with chemokine dimerisation[8]. Inspired by these exemplars, by using hydrogen-deuterium-exchange mass spectrometry (HDX-MS), we previously identified the chemokine-binding site of the tick class A evasin P672 (EV672_RHIPU) and developed a hexadecapeptide series that binds and inhibits the CC-class chemokines CCL8, CCL7, CCL3 and CCL2. We showed that one of these peptides inhibits inflammation in vivo[34]. In parallel, the chemokine-binding site of the tick class A evasin EVA4 was identified using NMR, and an EVA4-derived octadecapeptide Ev4Glu[14]-Asn[31] that inhibits CCL5 function thereby developed[28]. The finding that a short peptide sequence derived from an evasin could bind more than one chemokine raised the possibility that other promiscuous chemokine-binding peptides may be identified in the sequences of evasin proteins. The time-consuming and labour-intensive nature of HDX-MS or NMR approaches however render challenging their application to study the 386 pairwise evasin-chemokine interactions that have been documented to date[18–23,35].

Phage-display coupled to deep sequencing is a powerful approach that can be used to rapidly identify short linear interacting motifs (SLiMs) that mediate protein-protein interactions[36–39]. A phage-displayed peptide library derived from the first protein is selected using the second protein and bound phage are analysed using next-generation sequencing. Such approaches have focussed on identifying SLiMs from intrinsically disordered unstructured regions of proteins and have recently been applied to map domain-SLiM binding sites in the human proteome[39]. We hypothesised that phage-display selection coupled to deep sequencing, performed in parallel with multiple chemokines, could be exploited to identify promiscuous chemokine-binding peptides from evasins. Here we develop a pipeline combining phage-display, saturation mutagenesis and computational modelling to discover promiscuous chemokine-binding peptides from class A evasins and characterise the pharmacophore.

## Results

### Library construction and screening

We constructed a phage-display library where the major bacteriophage coat protein p8 was fused to hexadecapeptides derived from the mature sequences of 21 class A evasins[28,34] (see Fig. 1 and methods for details). This approach results in multivalent phage display, and allows identification of low affinity interactions[40]. We initiated our studies with class A evasins as linear chemokine-binding peptides had already been identified from this evasin class using structural methods[28,34], and would give us an opportunity to test the phage-display method. As unpaired Cys residues may compromise phage-display, they are mutated during library construction to alanine[38], a substitution that removes side-chains beyond the β-carbon without affecting conformational flexibility[41]. We constructed the library so that it included peptides with Cys residues intact, and also had peptides with mutations of Cys to Ala, and the conservative substitution, Cys to Ser[42]. Comparing counts in the wild-type Cys containing peptides, and those with Cys-Ser and Cys-Ala mutations obtained by next-generation sequencing of the input library show however that there was little or no impact on count numbers (Supplementary Fig. 1). The library was selected with 25 biotinylated chemokines as baits (Supplementary Table 1), individually attached to a

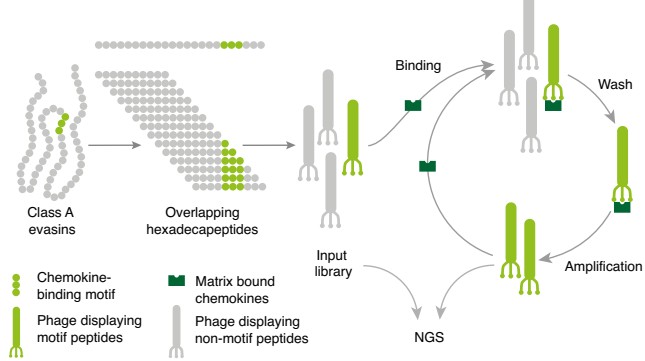

**Fig. 1 | Phage display selection of chemokine binding peptides.** Schematic of phage display nextGen sequencing pipeline. Class A evasins were deconstructed into DNA sequences encoding overlapping hexadecapeptides at single amino acid residue resolution and cloned into prSTOP4 phagemid vector. Phagemid pools were converted to phage library ('input') by transfecting *E.coli* together with helper phage. The phage-display library was amplified using PCR and the amplicon analysed using next-generation sequencing (NGS). Phage-display libraries were incubated with matrix-bound human chemokines and washed to remove any weak/non-binders. The matrix-bound phage was amplified using T1 phage-resistant *E. coli* and helper phage, and the amplified library was incubated again with the matrix-bound chemokine to further purify strong binders. Following three selection rounds the phage-inserts were amplified using PCR and the amplicon analysed using NGS. Hexadecapeptide enrichment (E) following selection was calculated as ratio of output peptide frequency to input peptide frequency and expressed as log2E. Peptides binding complement C5a (negative control) were excluded.

streptavidin matrix. Following library selection with each chemokine we calculated the enrichment (E) of each peptide in comparison to the input library, and expressed the enrichment as log2E, as this metric is correlated with binding affinity[43].

## Phage display identifies regions overlapping with known chemokine binding sites

Mapping each peptide to the parental sequence (Fig. 2) by log2E obtained for each chemokine identified overlapping hexadecapeptides that clustered in regions of the parental sequences. This is most evident for instance for EVA4, EV672, EV974 and EV546. To identify regions of the parental protein that contribute to chemokine-binding hexadecapeptides, we calculated the contribution of each residue (rlog2E) as the sum of log2Es of peptides overlapping a residue. We defined regions of interest where rlog2E exceeded the upper boundary of the 95% confidence interval of median rlog2E. The regions of interest overlap the known chemokine-binding sites for EVA4, EV672, and EV974 (Fig. 2). Surprisingly, we observed that hexadecapeptides from these regions, in addition to binding CC chemokines as expected, also bind CX3CL1 and several CXC chemokines (CXCL1, CXCL10, CXCL11, CXCL12B, CXCL13, CXCL5 and CXCL8) which are not known to bind the parental proteins.

## Phage-display identifies promiscuous chemokine binding peptides

To characterise peptides that bound chemokines promiscuously, we selected those that bound at least three chemokines with log2E > 5. We find that of 30 peptides thus identified (Supplementary Table 2), 18 that are wild-type have one or more Cys residues. In each case (i.e., EVA4, EV991, EV974, EV672, E1243, E1180) where disulfide bond data was available in UniProt or from Alphafold, this Cys residue was disulfide bonded in the parental evasin sequence. Nine of 12 Cys-mutant peptides are mutant versions of the 18 wild-type peptides. The heatmap of individual log2E values indicated that certain peptides (e.g., HD2, HD7) are highly promiscuous, and bind over 15 different chemokines including exemplars from CC, CXC and CX3C classes (Fig. 3a).

Interestingly, there is a clear distinction in the binding patterns of CXCL12 and its isoform CXCL12B. The cause for the differences may reside in the known effects of tag placement on CXCL12[44,45]. Additional residues present at the C-terminus of CXCL12B would change the distance of the biotin tag from the peptide-binding epitope and may alter the effect of the tag. The neighbour-joining cladogram (Fig. 3b) shows that EVA4, EV672, and the highly homologous evasins EV974 and EV546 contribute overlapping peptides.

## Sequence alignment reveals a conserved motif containing an unpaired Cys residue

The two largest groups of overlapping wild-type peptides are from EVA4 and EV672. Multiple sequence alignment of these peptides (Fig. 3c) identifies two linear motifs with conserved residues: E(E/D)(E/D)DY and P(L/V)TCYF. We next examined the importance of having an unmutated Cys residue by evaluating the total log2E for each peptide aggregated over all chemokines as a summary measure of binding affinity (Fig. 3d). This shows that wild-type peptides have significantly higher total log2E compared to mutant peptides i.e., where Cys had been mutated to either Ser or Ala. These results suggest that the unpaired Cys residue in the PL/VTCYF motif contributes to chemokine binding affinity.

## Biolayer interferometry confirms binding of HD2 to CC- and CXC- chemokines

We next evaluated the ability of the exemplar peptide HD2 to bind a panel of chemokines (Supplementary Table 3) using biolayer interferometry (BLI). We generated HD2 and a scrambled control (HD2SCR) as HIS:SUMO fusion proteins in *E.coli*. This approach results in soluble proteins that can be immobilised on to BLI sensors through the N-terminal HIS tag. In comparison to control, we observed that several CC and CXC-class chemokines (CCL1, CCL5, CCL7, CCL8, CCL11, CXCL10 and CXCL13) specifically and reproducibly bind to the immobilised HD2 peptide fusion (Supplementary Fig. 2). The binding of HD2 to these chemokines was characterised at different chemokine doses to evaluate binding affinity ($K_D$). We performed dose-response experiments for all chemokines in parallel and show that dose-dependent responses occur in each case (Fig. 4). We found that a 1:1 binding model was able to fit the data for CCL8 (Fig. 4a), allowing calculation of binding affinity (Supplementary Table 4). However, this binding model does not fit the data for the other chemokines, where we observed the signal accumulating over time at higher chemokine concentrations (Fig. 4b–h). As chemokines are known to oligomerize in solution and also upon glycosaminoglycan binding[46], the most parsimonious explanation for this binding profile is chemokine oligomerization at the BLI sensor. All chemokines bound in BLI were also bound by phage-displayed-HD2 where a phage-display experiment was performed. However, not all chemokines that bound HD2 in phage display were observed to bind by biolayer interferometry. The most likely explanation is that multivalent phage peptide-display allows identification of lower affinity interactions in comparison to BLI[40].

## HD2 inhibits both CC- and CXC- chemokines

We next studied the ability of the exemplar peptide HD2 to inhibit chemotaxis by CC and CXC chemokines (Fig. 5 and Supplementary Fig. 3). In comparison with a scrambled control (HD2SCR), we find that it significantly inhibits cell migration induced by the CC chemokines CCL3, CCL5, CCL7, CCL8, CXCL6, and CXCL10. Inhibition of migration in response to CCL2, CCL23 and CXCL11 was also observed, although this was not statistically significant. No inhibition was observed for CCL14 and 15. These results are generally concordant with the binding results obtained in phage-display and BLI except for CCL15, which bound in phage-display but did not inhibit cell migration. We did not test other peptides extensively, but where tested, observed inhibition of CCL8 by HD845 and HD540 (Supplementary Fig. 3a). We performed

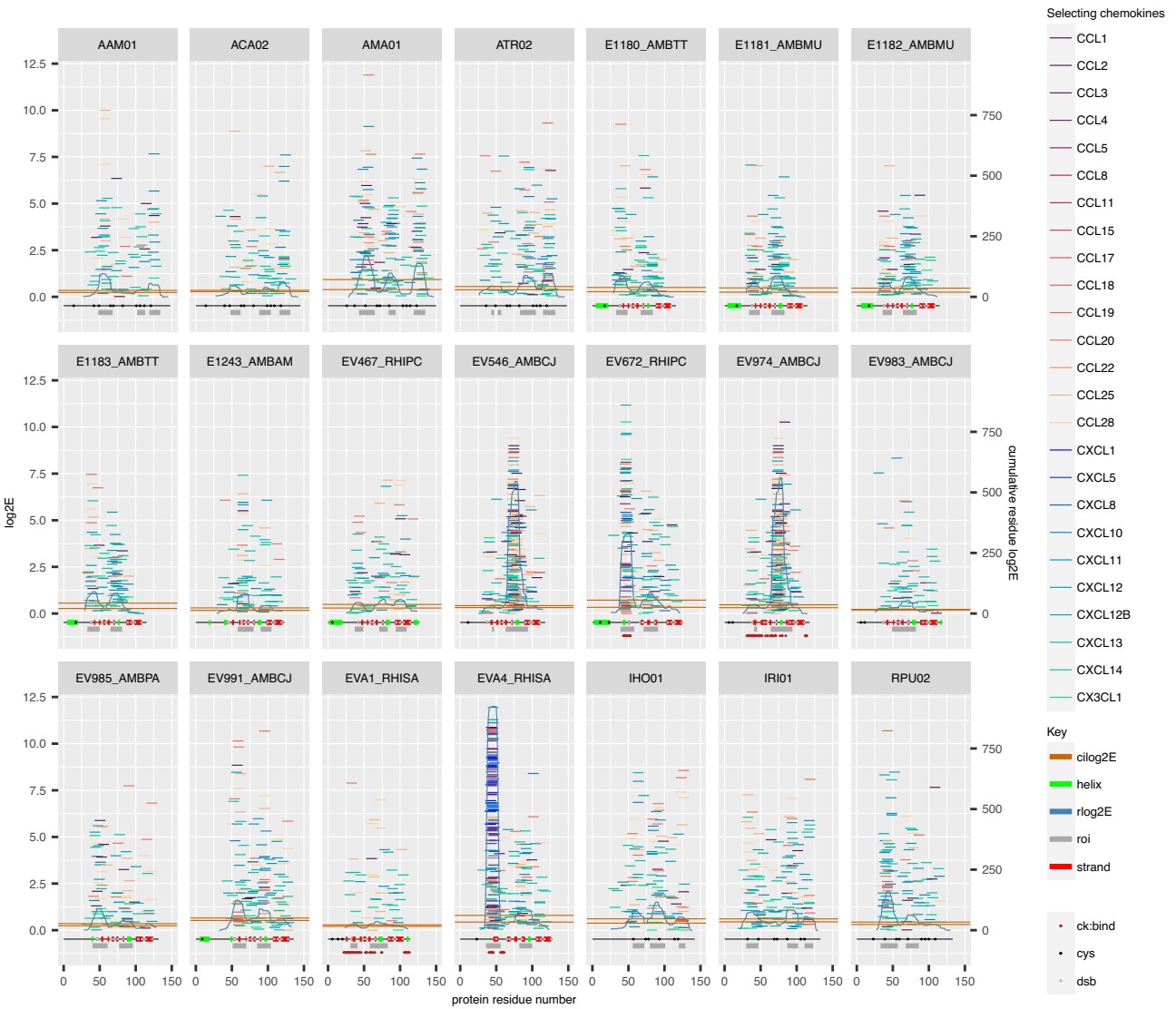

**Fig. 2 | Mapping of enriched hexadecapeptides to parental proteins.** Parental class A evasins are identified in the strip above each subpanel by UniProt entry names where available. Individual hexadecapeptides are represented as horizontal tiles, coloured by the selecting chemokine. Left *Y*-axis scale shows peptide log2E, and right *Y*-axis scale the residue log2E. Peptide log2E was calculated as described above. Residue log2E (rlog2E) for each residue was calculated as the sum of log2Es of peptides overlapping a residue, and the curve smoothed using a rolling median over a window of 7 residues. Regions of interest (roi) were identified where rlog2E exceeded the upper boundary of the 95% confidence interval of median rlog2E.

Structural features were obtained from the UniProt database which includes data from AlphaFold models. Known chemokine binding sites are indicated. Chemokine-binding sites for EV672 to CCL8 and EVA4 to CCL5 were obtained from published data[28,34]. Chemokine-binding sites on EVA1 for CCL3 were extracted from PDB structure 3FPU. Chemokine-binding sites on EV974 for CCL17 and CCL7, were extracted from PDB structures 7S4N and 7SOO[29]. ck:bind chemokine binding sites, cys cysteine, dsb disulfide-bonded cysteine, roi region of interest, cilog2E 95% confidence interval of median rlog2E for the entire protein. Source data are provided as a Source Data file.

dose-response experiments with HD2 against CCL8, CCL5, CCL7, CXCL10 and CXCL6 (Fig. 6). While EVA4 inhibits the CC chemokines (Fig. 6f–h), it did not inhibit CXCL10 or CXCL6 (Fig.6i, j), and dose response curves could not be obtained. The IC50 of the HD2 peptide ranged from a median 1.9E-8 molar (0.019 µM) for CCL8 to 1.4E-5 molar (14 µM) for CXCL6 (Fig. 6k).

**Alanine-scanning mutagenesis identifies contiguous residues in HD2 necessary for binding**

A molecular-level understanding of the binding mechanism is needed for the development of peptides as therapeutics and is usually obtained from structural analyses. As HD2 binds many chemokines this becomes a challenging task, and we explored if we could elucidate this using phage display mutagenesis. We examined the role of each residue in HD2 for binding in phage-display. We generated a library of HD2

mutations that had NNK substituted at each residue and performed phage-display selection against a panel of 24 biotinylated chemokines, in parallel, as described above. This strategy allowed us to evaluate the impact not only of Ala substitutions[41], but also conservative, hydrophilic and hydrophobic substitutions[47], by comparing with the binding of parental HD2, which was also included in the library. Ala substitution removes side chains beyond the β-carbon and can be used to infer the role of side-chain functional groups without affecting the conformational flexibility of the backbone[41]. Analysis following selection with the chemokine panel showed that several mutations have a large and significant impact on mean log2E when compared to parental HD2 (Fig. 7a). Contiguous regions of three or more residues that significantly reduce binding upon Ala substitution compared to parental HD2 are E2-Y5 and P10-Y14. These residues are within the two motifs identified previously. We next judged the impact of Ala mutation by

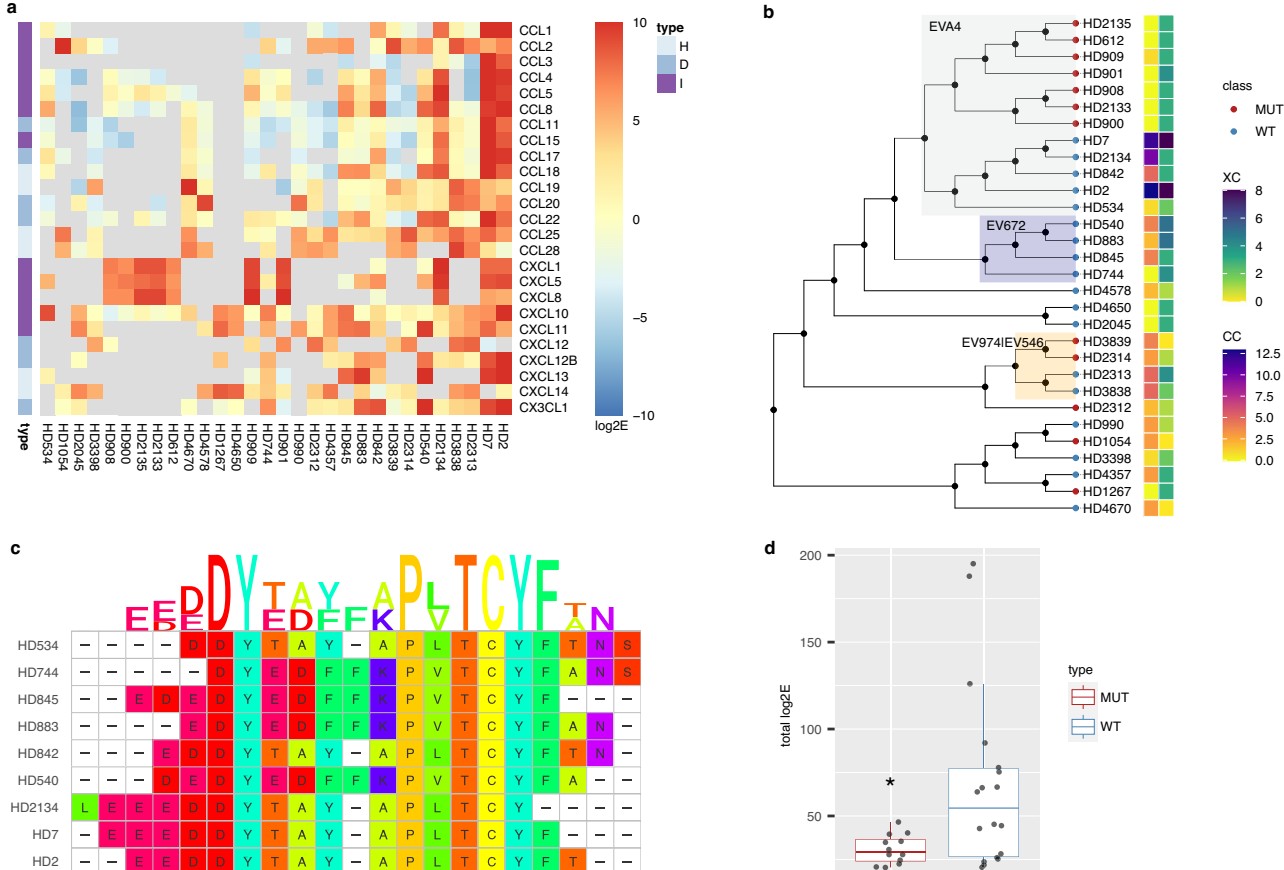

**Fig. 3 | Promiscuous peptides. a** Tileplot showing log2E of 30 peptides that specifically bind at least 3 chemokines with log2E > 5. Rows show the selecting chemokine and columns the peptide. Peptides are arranged by total numbers of chemokines bound with log2E > 5. Scale bar shows log2E values. Grey tiles indicates that the peptide was not recovered following the screen. Chemokine types (I = inflammatory, D = dual, H = homoeostatic) are indicated. **b** Neighbour-joining tree of peptides. Peptide identities are indicated at the tree tips. The ancestral node was defined by midpoint rooting. Peptides derived from EVA4, EV672 and EV974 are indicated in azure, navy, and orange respectively. The heatmap shows the number of CC or XC chemokines bound with log2E > 5. Mutant (MUT) and wild-type (WT) tip nodes are indicated. **c** Sequence alignment and logo for wild-type EVA4 and EV672-derived peptides. Amino acid residues are coloured using the Taylor scale. **d** Box-whisker plot showing the effect of Cys mutation to Ala or Ser (MUT) compared to wild-type (WT) for the subset of 30 peptides. Y-axis indicates the total log2E. Individual peptide data are indicated as points. The box-whisker plot shows the median as centre, 25th and 75th percentile as bounds, and 1.5*interquartile range as whiskers. A statistically significant difference between WT and MUT groups, using one-way ANOVA was observed, p = 0.0231, F = 5.782, df=1, n = 18 WT and 12 MUT independent peptide samples. Source data are provided as a Source Data file.

calculating the change in log2E (Δlog2E) between the mutant and parental HD2. A positive value of Δlog2E indicates increased binding, and a negative value decreased binding compared to parental HD2. A tile plot of the data (Fig. 7b) shows that while certain chemokines are relatively tolerant to single-point alanine mutagenesis (e.g., CCL8), others (e.g., CXCL10) were not. Analysis of conservative residue substitutions, identified from the Dayhoff PAM250 substitution matrix (Fig. 7c, Supplementary Fig. 4a) indicates that these substitutions have no significant deleterious effect on Δlog2E at the N-terminal motif, but that certain residues in the C-terminal motif (P10, T12 and C13) cannot be substituted. As may be seen, the log2E for the parental HD2 was -0 in these analyses indicating a lack of enrichment. Examination of our data shows that there are several mutations that enhance binding to chemokines (e.g., anionic mutations at T6 and A7 or hydrophobic mutations at L11, see Fig. 7f, and Supplementary Fig. 4) providing a likely explanation of this result as they would compete with the parental HD2 peptide for binding.

### Hydrophile-scanning mutagenesis identifies an N-terminal anionic patch in HD2

Ionic bonds are major interactions at protein interfaces, and occur between charged anionic and cationic residues[48]. Hydrophile scanning uses systematic mutation to anionic (e.g. glutamic or aspartic) or cationic (e.g. lysine, arginine) residues and complements alanine-scanning mutagenesis[47]. Analysis of these HD2 mutations present in the dataset generated above, shows that glutamic or aspartic acid (anionic) substitution at the N-terminus is not deleterious for binding, and in some cases, e.g., at T6, improve binding, whereas they are significantly deleterious at the C-terminus (Fig. 7d, Supplementary Figs. 4b, c, 5a, b). In contrast, cationic (lysine or arginine) mutations are generally deleterious through the peptide (Fig. 7e, and Supplementary Figs. 4d, e, 5c, d). Taken together these results indicate that an anionic N-terminal patch (EEDD) in HD2 mediates crucial interactions with target chemokines.

### Hydrophobe-scanning mutagenesis identifies a role for C-terminal hydrophobicity in HD2

Protein interfaces are frequently characterised by hydrophobic interactions, leading to exclusion of these residues from the water exposed surface[48]. We therefore asked whether systematic mutation to the hydrophobic residues (i.e, valine, isoleucine, methionine and leucine[49]) would allow us to identify HD2 residues that likely mediated hydrophobic interactions with chemokines. Analysis of these mutations, from the phage-display dataset generated above, showed that certain

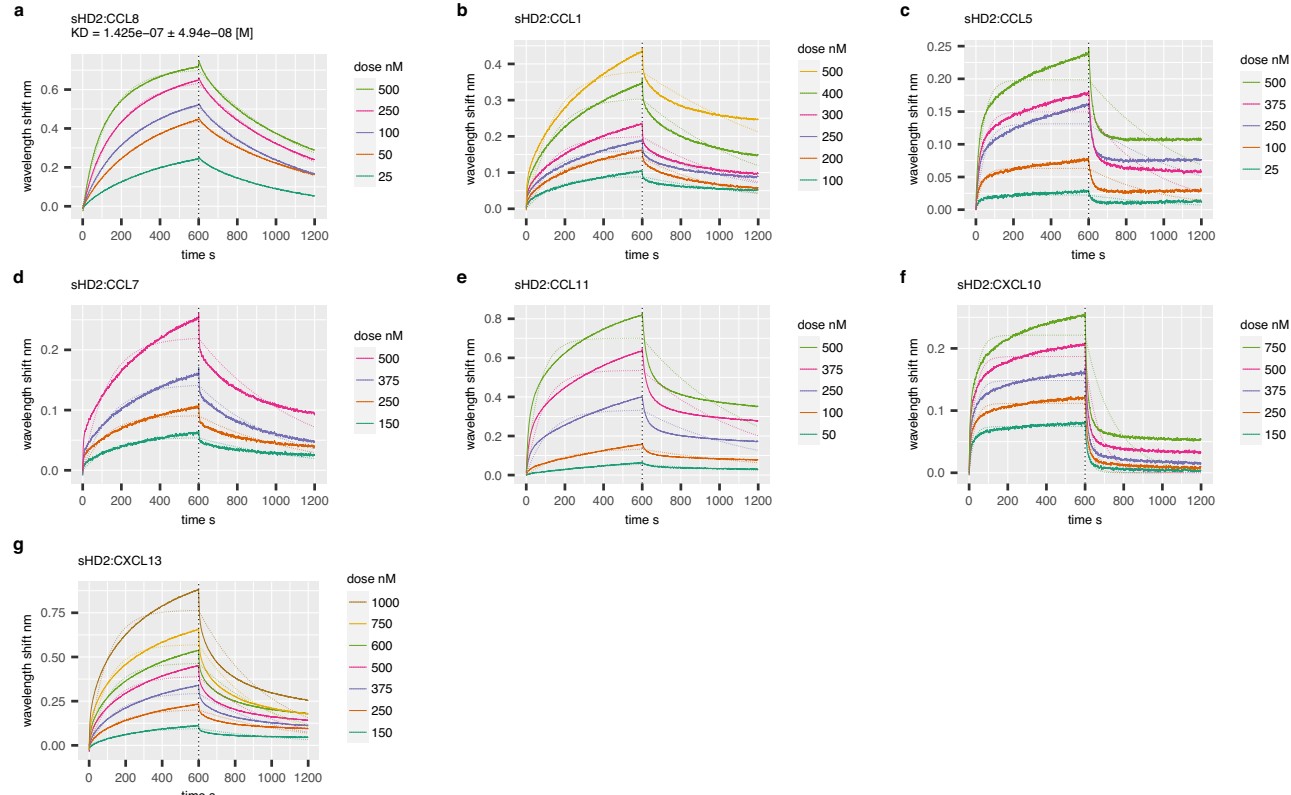

**Fig. 4 | Binding of chemokines using biolayer interferometry.**
**a**–**g** Representative BLI dose-response sensorgrams showing purified sHD2 (SUMO:HD2) binding to indicated doses of chemokines. Plots display wavelength shift (Y-axis; nm) versus time (X-axis; seconds). Vertical dotted line indicates the onset of dissociation. Raw data are indicated by solid lines and fitted data (using a 1:1 binding model) by dotted lines. $K_D$ estimates (mean ± standard error) are indicated where a good fit to the 1:1 binding model was obtained. Source data are provided as a Source Data file.

residues at the C-terminus—L11, T12, F15 and T16—can be substituted with a hydrophobic residue without significant deleterious effect (Fig. 7f and Supplementary Figs. 4f–i, 5e–h)).

## Alanine-scanning mutagenesis identifies functionally important residues in HD2

To determine if alanine substitutions impacted the ability of HD2 to inhibit chemokine function we performed chemotaxis assays with synthetic peptides. We examined the effect of HD2 alanine mutants on migrated cell counts in chemotaxis assays using chemokines CCL5, CCL7, CCL8, CXCL6 and CXCL10, and compared it to parental HD2 as control (Supplementary Fig. 6). We observed that while several mutations (at residues E1, E2, D4, Y5, Y8, P10, L11, C13, Y14, F15), significantly affect ability to inhibit chemotaxis (as evidenced by an increase in migrated cell count relative to parental HD2) by CCL5, CCL7, CCL8 and CXCL6 and CXCL10, only P10A significantly affects chemotaxis by CCL8. This data is summarised by chemokine in Fig. 8a, where the effect of residue mutated to change in migrated cell count is shown. We performed a meta-analysis of the alanine-mutant chemotaxis experiments by pooling all biological replicate datapoints to obtain an overview of residues critical for chemokine inhibition (Fig. 8b). This analysis revealed that key residues that significantly impact function over all chemokines when mutated to alanine are P10, L11, C13 and Y14. We next asked whether chemotaxis inhibition was correlated with binding in phage-display. We examined the relationship between change in migrated cell count (Δmigrated) with change in log2E (Δlog2E) for each peptide:chemokine pair where such data was available (Fig. 8c). We observed that these two properties are highly correlated. Mutant peptide:chemokine pairs with lower Δlog2E (i.e., poor binding) have higher Δmigrated values (i.e., poor

chemotaxis inhibition). Thus, poor binding in the phage-display experiment correlates with reduced inhibition of chemotaxis, indicating that the most likely explanation for loss of inhibition by a mutant is reduced binding affinity.

## HD2 binds CC and CXC chemokines at distinct locations

To understand how the peptide HD2 may bind and inhibit CC- and CXC- class chemokines we modelled the chemokine:HD2 complex using two different methods, AlphaFold2-Multimer[50] and AutoDock CrankPep[51] (Fig. 9, Supplementary Figs. 7, 8). For each model, five alternative docking poses were identified using AlphaFold2-Multimer and ten using AutoDock CrankPep. We calculated the Rosetta cross-interface binding energy for each docked structure (Supplementary Table 7) as this parameter shows highest AlphaFold model classification accuracy[52]. In all cases but one, the cross-interface binding energy was less than −16, the suggested cut-off value[52], supporting the docked models. To identify peptide-proximal regions on the chemokines, rather than analysing the single highest-ranked pose from each method, which is less likely to retrieve the native docking pose[53], we aggregated data from the docking poses into heat maps of weighted proximity scores (Fig. 9a, b, Supplementary Figs. 7–9). AlphaFold2-Multimer analysis suggested that HD2 is in proximity to the N-terminus, N-loop, and residues within the β3-strands, 30s and 40s loops of CCL2, CCL3, CCL5, CCL7 and CCL8, but in proximity to the β1-strand and α-helix of CXCL6, CXCL10 and CXCL11 (Fig. 9a–c, Supplementary Fig. 7). Analysis using AutoDock CrankPep indicated overall concordance with the AlphaFold2-Multimer results, except for CXCL11 and CXCL6, where the peptide is in proximity to the N-terminus (Supplementary Fig. 8). The results of HD2 (EEDDYTAYAPLTCYFT) docking using AlphaFold2-Multimer and AutoDock CrankPep are remarkably

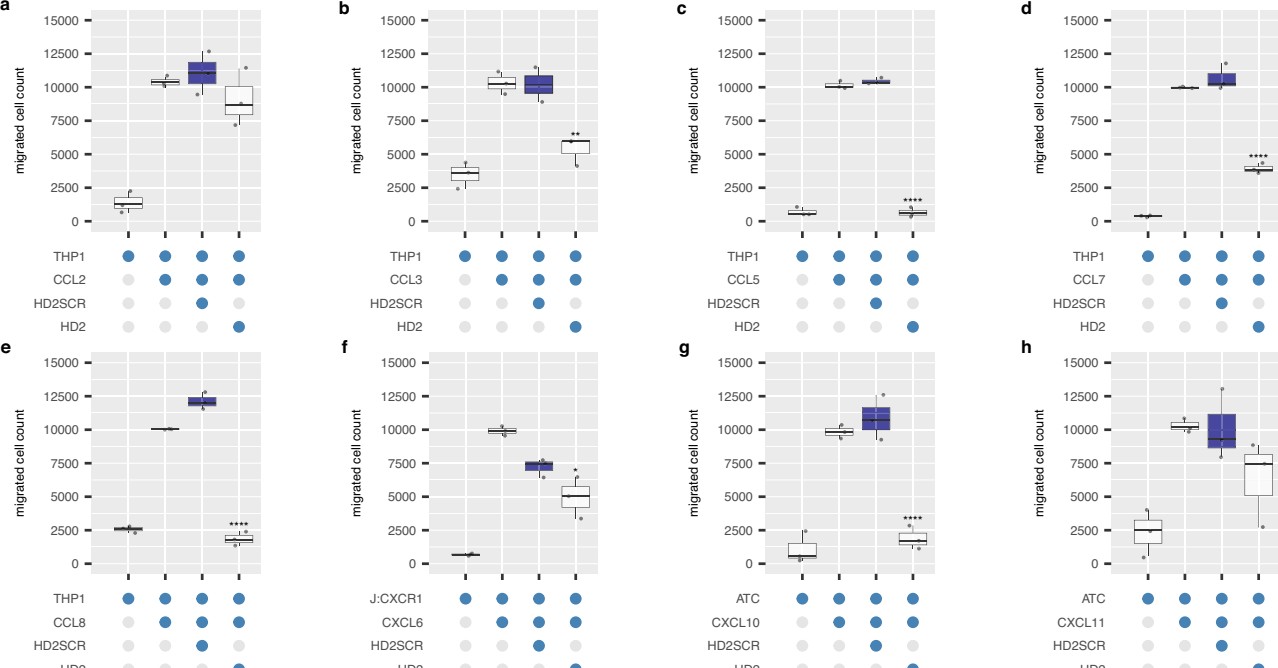

**Fig. 5 | Effect of exemplar peptide on chemotaxis. a–h** Box-whisker plots showing the effect of the exemplar peptide HD2 on cell migration induced by indicated human chemokines. All experiments were performed as three technical and three biological replicates, and individual biological replicate data points (mean of technical replicates) are shown. *Y*-axis in each panel shows cell count normalised to the median value of migrated cells in the presence of chemokine alone, set at 10000 cells. *X*-axis shows constituents of each experiment as blue-filled dots. Chemokine and cell type names are indicated. HD2SCR is a scrambled version of HD2 used as negative control. All peptides were at 10 µM final concentration and chemokines at EC80 doses. Each box-whisker plot shows the median as centre, 25th and 75th percentile as bounds, and 1.5*interquartile range as whiskers. Statistically significant differences (compared to control, coloured blue, *n* = 3 biologically independent experiments per group), were identified using a two-sided Dunnett's test with correction for multiple comparisons and are indicated by asterisks: ****$p \leq 0.0001$, ***$p \leq 0.001$, **$p \leq 0.01$. Exact *p*-values by panel are: (**b**) $p = 0.009456$; (**c**–**e**) $p = 0.000000$; (**f**) $p = 0.033149$; (**g**) $p = 0.000016$. Source data are provided as a Source Data file.

consistent with the previously reported NMR chemical shift perturbation model of a highly similar peptide Ev4Glu[14]-Asn[31] (EEEDDY-TAYAPLTAYFTN) in complex with CCL5[28] (Fig. 10), providing confidence in the docking methods we employed. AlphaFold2-Multimer models of EVA4 with target chemokines[35] suggested that it is in proximity to residues in the N-terminus and the N-loop, and residues within the 30s and 40s loops of CCL2, CCL3, CCL5, CCL7 and CCL8 (Supplementary Fig. 9). These results are also consistent with NMR analysis of CCL5:EVA4 interactions reported previously[28]. The HD2 segment within the chemokine:EVA4 models occupies a similar position to that observed in the corresponding chemokine:HD2 models (Supplementary Fig. 9a–e), and the chemokine residues in proximity to EVA4 in these models are similar to those in proximity to HD2 (Supplementary Fig. 9g).

### HD2 binding partially occludes receptor-binding sites
As models of chemokines with their receptors are not uniformly available, we also modelled chemokine:receptor complexes using AlphaFold2-Multimer (Fig. 9, Supplementary Figs. 7, 9), to identify the receptor-proximal regions of these chemokines. These analyses suggest that CCR1, CCR2, and CCR5 are in proximity to the N-terminus, N-loop, β1 and β3-strands and residues within the 30s and 40s loops, while CXCR3 and CXCR1 are also in proximity to residues in the α-helix respectively (Fig. 9a–c, Supplementary Fig. 7a–f). These analyses also show that the distal N-termini of the CXC-chemokine receptors CXCR3 and CXCR1 are predicted to bind to the β1-strands of cognate CXC-chemokines by wrapping around the chemokine, and are consistent with reported models of CC and CXC-chemokine interactions with chemokine receptors[54,55]. The overlap between peptide-proximal and receptor-proximal regions (summarised in Fig. 9c) suggests that the

peptide likely functions by partially occluding the receptor-proximal regions, interfering with binding. Analysis of the average number of chemokine bonds from the AlphaFold2-Multimer models for each HD2 residue indicated that the largest numbers of interactions, on average, are formed by the sequences E1:Y5 and L11:F15, i.e., the sequences EEDDY and LTCYF (Fig. 9d), with the unpaired Cys residue itself contributing many interactions.

## Discussion
In this study we performed phage-display selection in parallel with multiple chemokines, coupled to deep sequencing, to systematically identify overlapping hexadecapeptides from 21 biochemically characterised class A evasins that bind chemokines. By pooling the results from 25 independent experiments conducted with CC- and CXC-class chemokines we were able to identify regions of interest within the parental proteins that bound chemokines in a "one-to-many" manner. These regions of interest map to interaction sites identified previously by hydrogen-deuterium-exchange mass spectrometry of the EV672:CCL8 complex[34], by NMR for EVA4:CCL5 complex[28], and by X-ray crystallography for EV974 complexes with CCL17 and CCL7[29].

A surprising and important result of our study is that several peptides—unlike the parental class A evasin proteins from which they are derived[25]—bind, and in the case of the peptide HD2, inhibit both CC and CXC class chemokines. The peptide HD2 is from the same region of EVA4 as an octadecapeptide Ev4Glu[14]-Asn[31] which was synthesised based on NMR analysis of the EVA4:CCL5 interface[28]. Unlike HD2, the octadecapeptide has a Cys-Ala mutation, and was reported to inhibit a single chemokine, CCL5[28]. As shown in the present study, the example peptide HD2 also shares sequence homology with several EV672-derived peptides, including HD845. HD845 was previously described

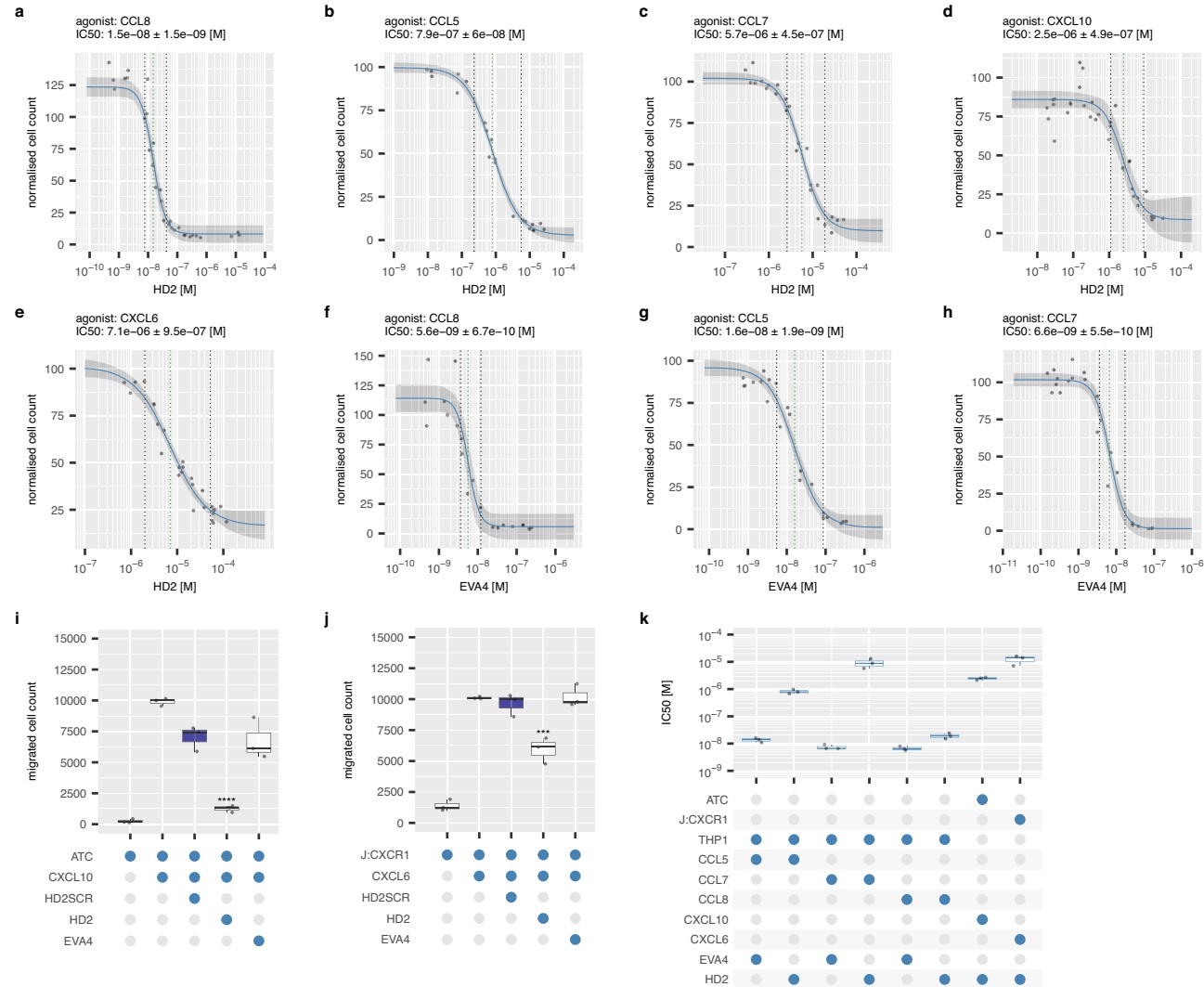

**Fig. 6 | Effect of HD2 and EVA4 on chemotaxis response. a–h** Representative dose-response curves showing effect of human chemokine induced THP1 or activated T cell (ATC) or Jurkat CXCR1 (J:CXCR1) migration by HD2 or EVA4. *Y*-axis shows percent migrated cells normalised to chemokine alone (set at 100%). *X*-axis shows inhibitor concentration (molar). Technical replicates at each inhibitor concentration (*n* = 3) are shown as individual data points. The dose-response curves (solid blue lines) and 95% confidence intervals (grey ribbons) were calculated using a 4-parameter log-logistic plot. Dotted green lines indicate IC$_{50}$ and dotted black lines IC$_{20}$ and IC$_{80}$. The IC$_{50}$ (M, estimated from the *X*-intercept) ± standard error of the estimate is indicated in each plot. **i**, **j** Box-whisker plots showing the effect of HD2, scrambled version HD2SCR, and parental evasin EVA4 on cell migration induced by CXCL10 and CXCL6 respectively. All experiments were performed as three technical and three biological replicates, and individual biological replicate data points (mean of technical replicates) are shown. HD2, HD2SCR and EVA4 were

at 20 μM final concentration in (**i**) and 10 μM in (**j**). *Y*-axis shows cell count normalised to the median value of migrated cells in the presence of chemokine alone, set at 10000 cells. *X*-axis shows constituents of each experiment as blue-filled dots. Each box-whisker plot shows the median as centre, 25th and 75th percentile as bounds, and 1.5*interquartile range as whiskers. Statistically significant differences (compared to control, coloured blue, *n* = 3 biologically independent experiments per group), were identified using a two-sided Dunnett's test with correction for multiple comparisons and are indicated by asterisks: ****$p \le 0.0001$, ***$p \le 0.001$. Exact *p*-values by panel are: (**i**) $p = 0.000020$; (**j**) $p = 0.000556$. (**k**) Summary IC$_{50}$ values for inhibition by HD2 peptide and parental evasin EVA4. *Y*-axis shows IC50 (molar) and *X*-axis the constituents of each experiment. Data is shown as a box-whisker plot with median as centre, 25th and 75th percentile as bounds, and 1.5*interquartile range as whiskers, of biological replicates (*n* = 3) shown as individual data points. Source data are provided as a Source Data file.

as BK1.5, and together with other peptides of the BK series, were designed based on HDX-MS analysis of the EV672:CCL8 interface[34]. None of the previously described class A evasin-derived peptides were shown to have anti-chemokine activity against CXC-class chemokines. Potential explanations for the lack of CXC-binding capacity of the parental evasin is that CXC-chemokine interacting residues on the peptide may be unavailable in the parental protein. For instance, the unpaired Cys residue in the peptide is invariably disulfide bonded in the parental evasin. Multiple lines of evidence—including sequence conservation in binding peptides, mutation of Cys to Ala or Ser resulting in loss of binding activity in phage display, and loss of functional activity in chemotaxis assays—suggest that the unpaired Cys

residue in these peptides is important. Cys residues are known to have non-covalent interactions[56], and it is possible that these enhance chemokine binding and inhibition. This result is consistent with the loss of activity previously observed with a Cys-Ala mutation in the EV672-derived peptide BK1.5 (which is identical to HD845 reported here)[34].

Following the discovery of the peptides by phage-display, we demonstrated that not only does the exemplar peptide HD2 bind multiple CC and CXC-chemokines using an independent approach—biolayer interferometry—but that it also inhibits multiple CC and CXC-chemokines in chemotaxis assays. These results suggest that the broad-spectrum chemokine-inhibiting activities of HD2 could be applied for therapeutic development. A key requirement for such

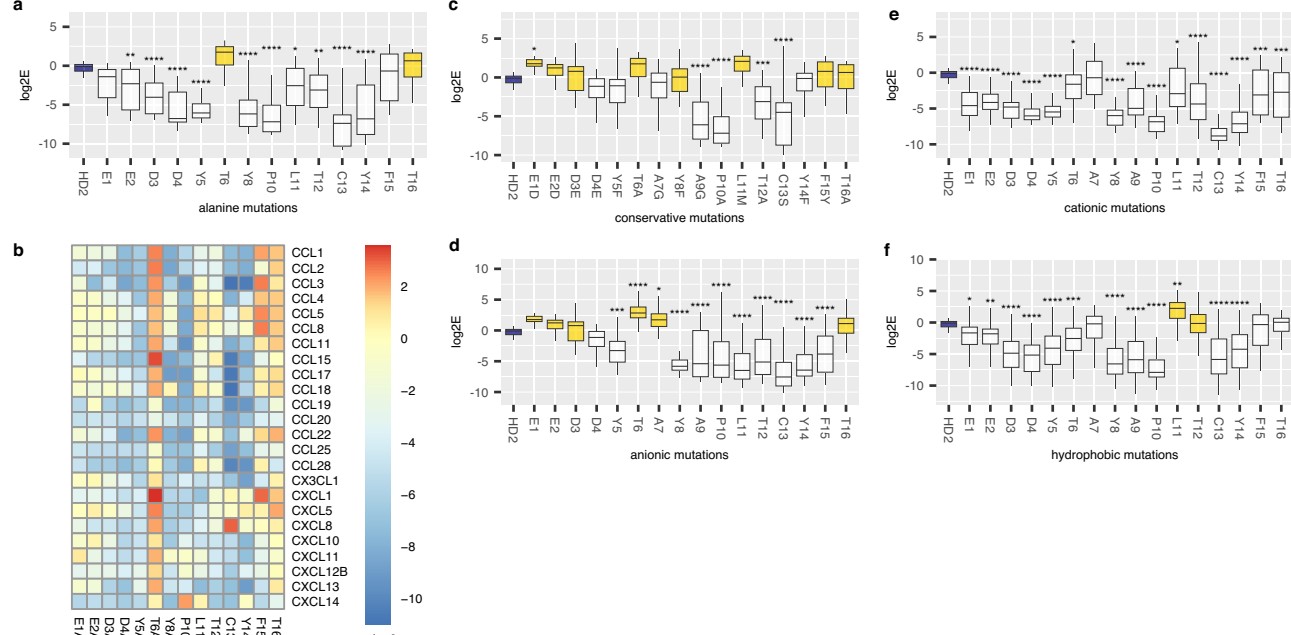

**Fig. 7 | Effect of HD2 mutations on phage binding.** Box-whisker plots (**a**, **c**–**f**) showing impact of HD2 residue mutation to alanine, conservative, anionic (glutamic acid, aspartic acid), cationic (lysine, arginine), and hydrophobic (leucine, isoleucine, methionine, and valine) residues respectively (*X*-axis) upon log2E (*Y*-axis). Each box-whisker plot shows the median as centre, 25th and 75th percentile as bounds, and 1.5*interquartile range as whiskers. Statistically significant differences (compared to control, coloured blue), using Dunnett's test with correction for multiple comparisons, are indicated by asterisks: ****$p \leq 0.0001$, ***$p \leq 0.001$, **$p \leq 0.01$, *$p \leq 0.05$, $n = 48$ per group for anionic or cationic mutations, 24 for wild-type, alanine, and conservative mutations, and 96 per group for hydrophobic mutations, except L11 where it was 72. Exact *p*-values are provided in Supplementary Table 5. Boxes showing a positive value for difference from control (identified from a two-sided Dunnett's test) are coloured yellow. **b** Tile plot of HD2 alanine mutations with tile colour showing Δlog2E, which is the difference in log2E between the parental wild-type peptide and the mutant variant following phage-display selection. Rows show the selecting chemokine and columns the mutation. Scale bar shows Δlog2E values. Source data are provided as a Source Data file.

progression, for instance in computer-aided rational drug design, is an understanding of the pharmacophore, i.e., the electronic and steric features required for binding and functional inhibition of the target[57,58]. Sequence conservation across peptides derived from EVA4 and EV672 that bound both CC and CXC chemokines indicated the presence of conserved residues in two motifs, E(E/D)(E/D)DY at the N-terminus; and (L/V)TCYF at the C-terminus. We applied saturation mutagenesis phage-display to dissect the functional importance of these residues. By examining the binding characteristics of alanine, hydrophilic and hydrophobic mutants for 24 different CC and CXC-chemokines, we show that the two motifs are biochemically distinctive, with the former characterised by an anionic patch and the latter by hydrophobic, aromatic and cysteine residues. These types of residues are some of the major drivers of protein complex formation[48]. We confirmed, using alanine scanning mutagenesis, the relative importance of these residues in functionally inhibiting multiple CC and CXC-chemokines and showed that functional inhibition strikingly correlates with binding in phage-display. The two motifs are separated by a conserved Pro residue which, while not contributing significantly to inter-chain bond formation, appears to be important for binding and inhibition by HD2. This is consistent with the role of the Pro residue in the analogous peptide BK1.1 (which is HD845 with a Cys-Ala mutation) observed previously[34], and suggests that it plays a role in peptide conformation, a known function of Pro residues[59].

To further elucidate the pharmacophore, we complemented these biochemical studies with two different in silico approaches that do not require prior knowledge of the binding pose of the peptide and use substantially different computational approaches. We initially applied AlphaFold2-Multimer[50], which is an implementation of the AlphaFold2[60,61] machine-learning algorithm "trained for multimeric inputs with known stoichiometry", and has been used for

peptide:protein docking[62]. In these studies, we used a 1:1 stoichiometry based on our previous observations that the predominant binding stoichiometry of BK1.1 to CCL8 is 1:1[34]. We found using AlphaFold2-Multimer that the exemplar peptide, HD2, is predicted to bind to distinct sites in CC and CXC-chemokines: i.e., in CC to the N-terminus, and in CXC to the first β-strand. These results were confirmed in the large part using AutoDock CrankPep[51,63] which folds flexible peptides and docks them into rigid targets[51,63]. The reason for only partial and not full concordance between the two methods may lie in the differences in the folding algorithms, and how they rank binding modes. Our analysis shows that HD2 binding sites in both CC and in CXC-chemokines overlaps with the receptor-binding regions of the chemokine—i.e., the sites predicted by AlphaFold2-Multimer to bind the chemokine receptor—suggesting that HD2 occludes distinct receptor-binding regions in CC and in CXC-chemokines, interfering with their binding to cognate receptors. Confidence in our docking approach is increased by the observation that the NMR-based model[28] of CCL5 with Ev4Glu[14]-Asn[31] (which differs from HD2 by three residues) is highly consistent with the models of HD2 docked to CCL5 produced by AlphaFold2-Multimer and AutoDock CrankPep. Using Arpeggio[64], an algorithm that calculates interactions at protein-protein interfaces, we show that the N-terminal motif of HD2 contributes ionic and the C-terminal motif hydrophobic interactions, consistent with the results of the anionic, cationic, and hydrophobic substitution experiments. A limitation of these studies is that we have not confirmed the in silico predicted interactions and structure-activity relationships using approaches such as NMR or X-ray crystallography, which need to be performed in the future.

In summary, we have identified, using phage display and deep sequencing, a series of peptides from class A evasins that display broad-chemokine binding and inhibition activity spanning both CC

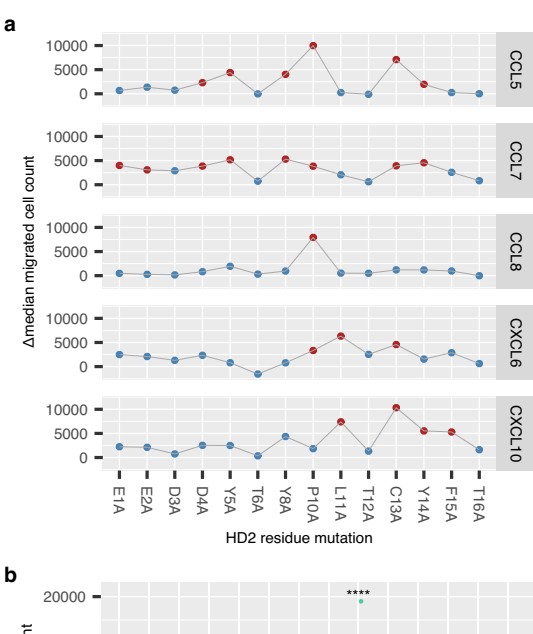

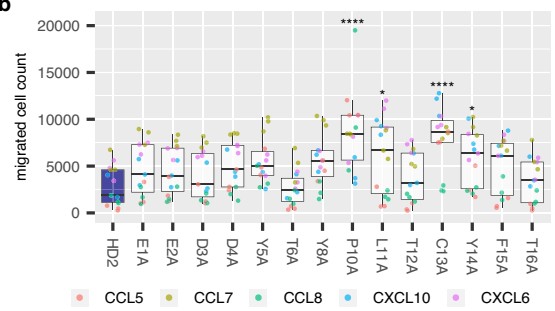

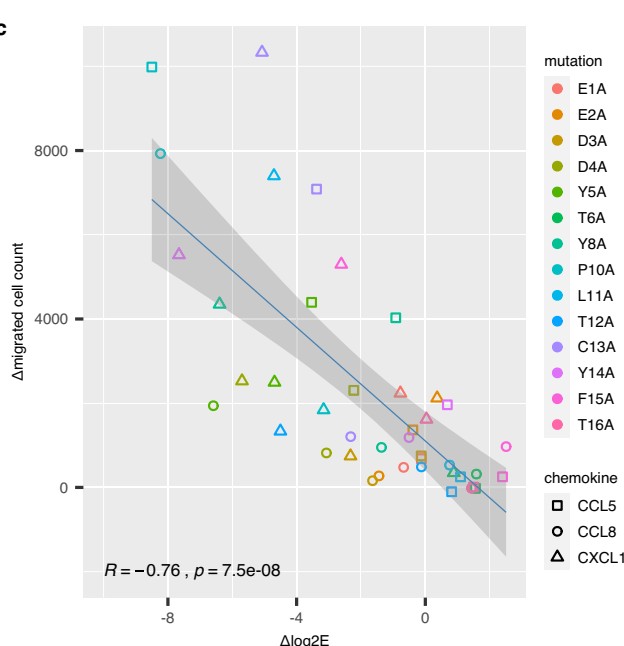

**Fig. 8 | Effect of HD2 alanine mutations on chemotaxis. a** Faceted plots of change in median migrated cell count between alanine mutant and wild-type HD2 peptide obtained in cell migration assays (Y-axis, Δmedian migrated cell count) versus alanine mutated residue (X-axis). Chemokines used for each experiment are indicated in the strip to the right of each plot. Statistically significant differences compared to parental HD2 are shown as red dots. **b** Box-whisker plot showing impact of HD2 residue mutation to alanine (X-axis) upon migrated cell count (Y-axis) for all chemokines studied. All experiments were performed as three biological replicates, and individual data points are indicated, and coloured by the chemokine used. The box-whisker plot shows the median as centre, 25th and 75th percentile as bounds, and 1.5*interquartile range as whiskers. Statistically significant differences (compared to control, coloured blue), using Dunnett's test with correction for multiple comparisons, are indicated by asterisks: ****$p \le 0.0001$, ***$p \le 0.001$, **$p \le 0.01$, *$p \le 0.05$, $n = 15$ observations per group (three biological replicates for each chemokine). Exact $p$-values are: C13A 1.62e−05; L11A 2.86e−02; P10A 2.79e−06; Y14A 4.17e−02. **c** Correlation of phage binding and inhibitory potency for alanine mutant HD2 peptides. Scatterplot of Δmedian migrated cell count (Y-axis, obtained in cell migration assays) versus Δlog2E (X-axis, obtained in phage-display experiments) for peptide:chemokine pairs. Individual data points indicate peptide chemokine pairs for which both cell migration and Δlog2E were available. Linear regression plot with 95% confidence interval, Spearman correlation coefficient ($R$) and statistical significance ($p$) are shown. Source data are provided as a Source Data file.

physicochemical properties of parental peptides but overcome their pharmacokinetic limitations[65]. Such synthetic agents may be generated using unnatural or D-amino acids so that key interactions with the target are maintained. The binding affinity and inhibitory potency of the peptides described are not yet in the range that would be desirable for use as therapeutic agents. These parameters could be enhanced by directed in vitro molecular evolution using deep mutational scanning combined with phage-display[66]. Further enhancement of inhibitory potency is also possible by conjugation to immunoglobulin Fc-chains to increase target avidity[67]. We suggest that the peptides and pharmacophore identified in this study could form templates for broad-spectrum anti-chemokine peptibodies or peptidomimetics that could be developed as therapeutics to target inflammatory disease. The phage-display, saturation mutagenesis and computational docking pipeline developed here could provide a new route to the discovery of such therapeutics that disrupt redundant and hence resilient protein-protein interaction networks in immuno-inflammation and cancer[9,68].

## Methods

### Human blood samples

Peripheral blood cells were obtained from anonymized donor leuco-cyte cones purchased from NHS Blood Transfusion Services, with ethics approval obtained from the University of Oxford Medical Sciences Interdivisional Research Ethics Committee, CUREC1 approval reference R75963/RE001.

### Chemokines

Sources of chemokines for phage-display experiments, BLI, and cell migration are provided in Supplementary Tables 1 and 3.

### Plasmids

HIS:SUMO peptides were constructed using NEB Builder HiFi Assembly (E5520, NEB) with a plasmid backbone derived from HIS6:SUMO:CCL8[22]. The CXCR1-expression plasmid (D1398) expresses a human CXCR1-IRES2-blasticidin resistance cassette from a CMV-T7 promoter and was constructed using GoldenGate Assembly from idempotent parts[20]. The EVA4 expression plasmid P1922 (8xHis-EVA4_RHISA) was constructed in plasmid pHLSec[69] using an In-Fusion cloning kit (Takara) following the instructions of the manufacturer, and contains EVA4_RHISA mature peptide (i.e., residues 24–127) with a N-terminal His-tag. Plasmid sequences were confirmed by Sanger sequencing (Azenta/Genewiz UK).

and CXC-class chemokines. The pharmacophore mediating this activity was characterised using a combination of sequence conservation across peptides recovered, analysis of alanine, hydrophilic and hydrophobic mutants, and integration with computational modelling. Our analyses suggest that these peptides occlude different parts of the chemokine receptor-binding sites and contain two motifs, one with an anionic patch primarily contributing ionic interactions, and the second containing an essential Cys residue and contributing hydrophobic interactions. The pharmacophoric features thus identified could be exploited to develop shorter linear peptides, cyclic peptides and peptidomimetics—i.e. synthetic agents that mimic the 3D-spatial and

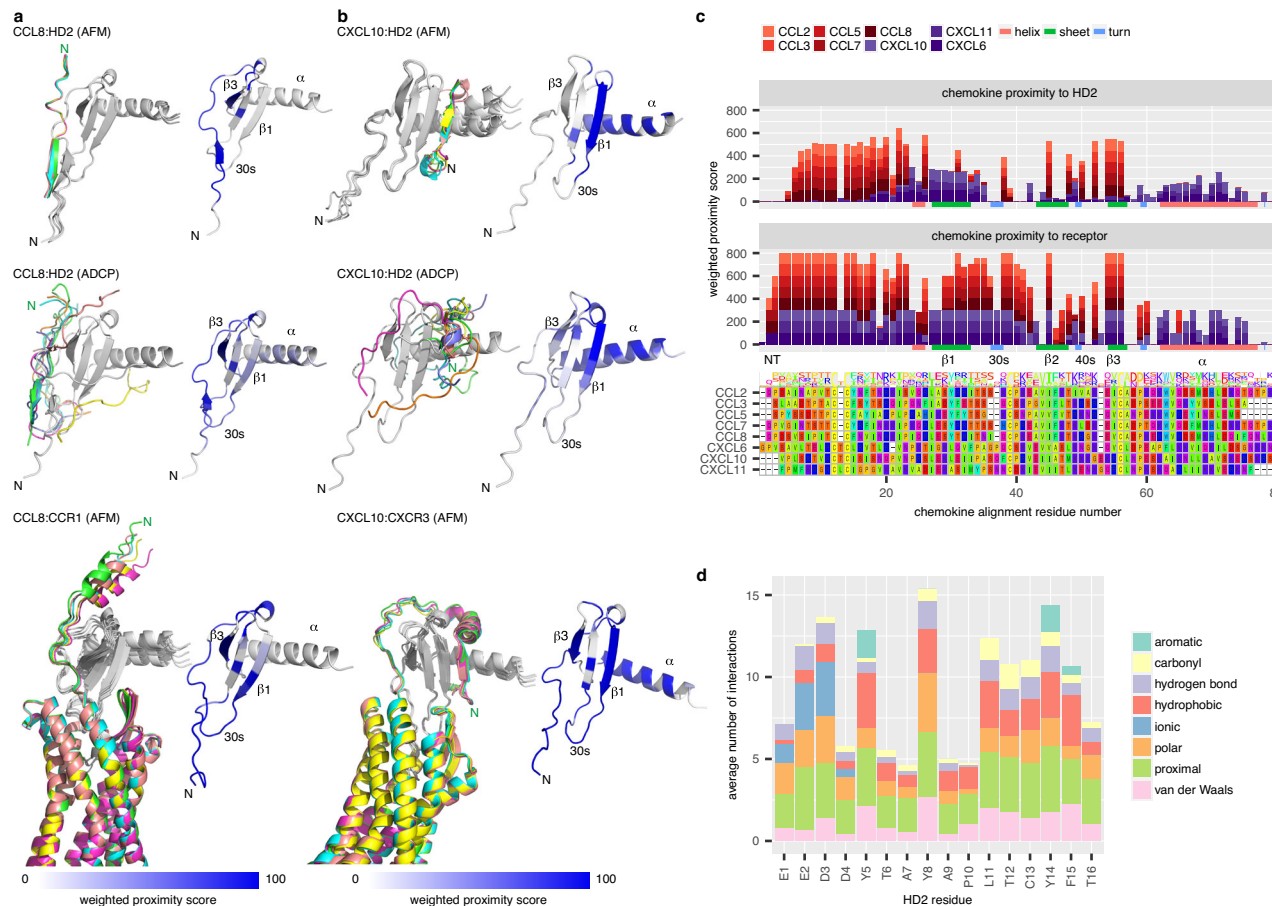

**Fig. 9 | Predicted binding modes of HD2 and receptors to chemokines. a** *Left panels:* Ribbon diagrams showing predicted poses for CCL8:HD2 using AlphaFold2-Multimer (*top*) or AutoDock CrankPep (*middle*) and for CCL8:CCR1 using AlphaFold2-Multimer (*bottom*). Chemokines are in grey, and the peptide or receptor in colour, with the top-ranked pose coloured green. *Right panels:* Corresponding heatmaps of weighted proximity scores mapped onto ribbon diagrams of CCL8. **b** *Left panels:* Ribbon diagrams showing predicted poses for CXCL10:HD2 using AlphaFold2-Multimer (*top*) or AutoDock CrankPep (*middle*) and for CXCL10:CXCR3 using AlphaFold2-Multimer (*bottom*). *Right panels:* Corresponding heatmaps of weighted proximity scores mapped onto ribbon diagrams of CXCL10. **c** Stacked bar charts of weighted proximity scores (*Y*-axis) for HD2 (*top panel*) and for receptor (*middle panel*), plotted for each chemokine residue (*X*-axis) following multiple sequence alignment (*bottom panel*). Models of chemokine:HD2 and

chemokine:receptor (CCL2:CCR2, CCL3:CCR1, CCL5:CCR5, CCL7:CCR1, CCL8:CCR1, CXCL10:CXCR3, CXCL11:CXCR3 and CXCL6:CXCR1) were generated using AlphaFold2-Multimer. Secondary structural elements for CCL8 are indicated at the bottom of the top and middle panels. Chemokines and secondary structural elements are coloured as indicated in the legend. Amino acid residues are coloured using the Taylor scale. **d** Stacked bar chart of chemokine:HD2 interchain interactions identified by Arpeggio from AlphaFold2-Multimer predictions. *X*-axis shows HD2 peptide residue and *Y*-axis the average number of interactions per residue. Interaction types are coloured as shown in the legend; "proximal" indicates residues within 5 Å of the chemokine chain. NMR nuclear magnetic resonance, AFM AlphaFold2-Multimer, ADCP AutoDock CrankPep, N N-terminus of the top-ranked pose, β1, β3 β-strands, α: α-helix. Source data are provided as a Source Data file.

## Cell lines

Jurkat E6.1 cells (ATCC TIB-152) were a gift from Pauline van Diemen (Oxford). THP1 cells (ECACC 88081201) were obtained from ECACC. Cell lines were confirmed mycoplasma free by monthly testing using a MycoAlert™ kit (LT07-118, Lonza) following the manufacturer's instructions. The J:CXCR1 cell line was generated by transfecting Jurkat E6.1 cells by electroporation with PvuI-linearised plasmid D1398 and selecting in RPMI-1640 (R0883, Sigma), 10% FBS (F9665, Sigma), 5 mM L-Glutamine (G7513, Sigma), and 5 µg/mL blasticidin (203350, Sigma). T-cells were isolated, activated and expanded as follows. Peripheral blood mononuclear cells were isolated from human leucocyte cones by density gradient centrifugation using Lymphoprep™ density gradient medium (07801, STEMCELL Technologies) and a SepMate™ isolation tube (85450, STEMCELL Technologies). CD8+ T-cells were isolated using two rounds of the human CD8+ T-cell isolation kit (480011, BioLegend). Isolated CD8+ T-cells were activated in ImmunoCult™-XF T-cell expansion medium (10982, STEMCELL Technologies) with an anti-CD3/anti-CD28 T-cell activator at 25 µl/ml (10991, STEMCELL Technologies) and supplemented with 10 ng/ml

recombinant human IL-2 (200-02, PeproTech) with 1% penicillin/streptomycin. During activation, T-cells were passaged every 2–3 days in TexMACS medium (130-097-196, Miltenyi Biotec), supplemented with 10 ng/ml human IL-2 (200-05-250, Peprotech) and incubated at 37 °C in 5% $CO_2$. At day 10-13 after initial activation, CD8+ T-cells were frozen at $20 \times 10^6$ cell/ml in TexMACS medium supplemented with 10% DMSO (D2650, Sigma) in liquid nitrogen. 24 h prior to use, activated CD8+ T-cells were recovered in TexMACs medium at $0.3 \times 10^6$ cell/ml and incubated at 37 °C in 5% $CO_2$.

## Phage display library design

Wild-type class A evasin nucleotide sequences (excluding the signal peptide encoding sequence) were codon-optimised for *E.coli* expression using GeneDesigner, using default settings (i.e. codon bias threshold 0.1, and removing splicing, RNA destabilising, prokaryotic ribosome binding site, Shine-Dalgarno sequences, optimising the 5′ structure, and removing repeats). Codon optimisation was also repeated after mutating each Cys residue to Ala and Ser. We designed 81-mer oligonucleotides such that they encoded hexadecapeptides

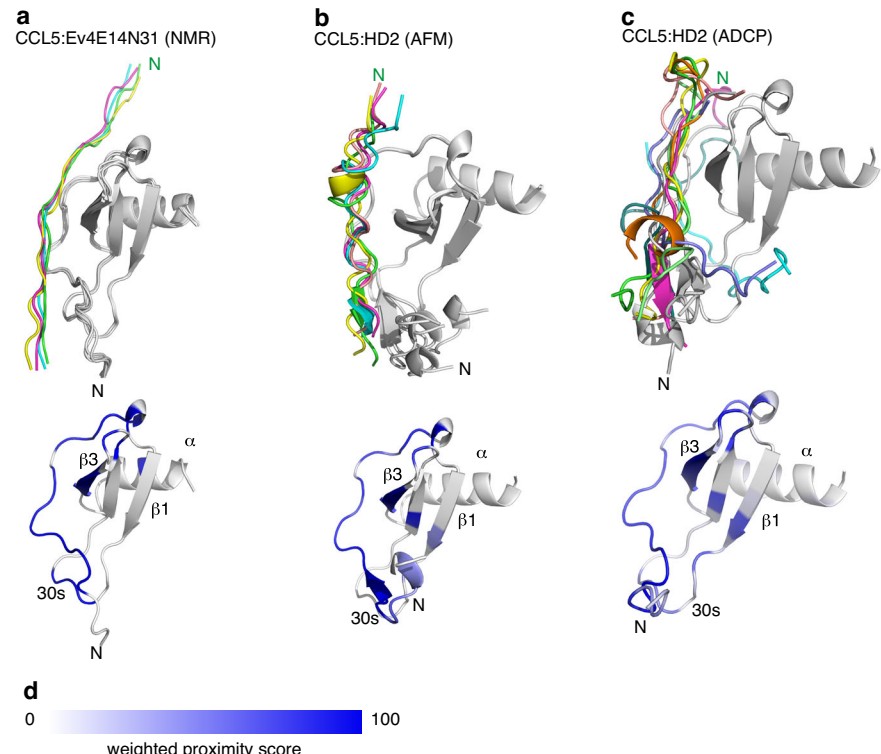

**Fig. 10 | Comparison of NMR-predicted binding to predictions by AlphaFold2-Multimer and AutoDock CrankPep. a–c** *Top panels:* Ribbon diagrams showing the predicted poses of indicated peptides bound to CCL5. Chemokines are in grey, and the peptide poses in colour, with the top-ranked pose coloured green. *Bottom panels:* Corresponding heatmaps of weighted proximity scores mapped onto ribbon diagrams of CCL5. Chemokine structural features (α-helix, β-strands and 30s loop) are indicated. **a** Poses obtained following the previously reported HADDOCK[76] analysis of NMR chemical shift perturbation data of peptide Ev4E14N31 (Ev4Glu[14]-Asn[31], EEEDDYTAYAPLTAYFTN) in complex with CCL5[28]. **b, c** Poses obtained following AlphaFold2-Multimer (AFM) and AutoDock CrankPep (ADCP) analysis respectively of peptide HD2 (EEDDYTAYAPLTCYFT) docked to CCL5. **d** Heatmap scale for weighted proximity score. NMR nuclear magnetic resonance, AFM AlphaFold2-Multimer, ADCP AutoDock CrankPep, N N-terminus of the top-ranked pose, β1, β3: β-strands, α α-helix. Source data are provided as a Source Data file.

overlapping by a single residue, and had the sequence 5′-GCAGCCTCTTCATCTGGC, and GGTGGAGGATCCGGA-3′ at respective ends to enable amplification and cloning. A total of 4741 distinct peptides (including peptides where Cys was mutated to Ala or Ser) were designed as oligonucleotides and synthesised as a pool (Genscript, 12 K chip).

## Phage display library construction
The plasmid prSTOP4 (kind gift from Dr Sachdev Sidhu, University of Toronto) was modified to have an *NsiI* restriction site and was amplified using primers (Sigma):

pRSTOP4Nsi_fwd: 5′-GGAGGCGCCGAGGGTGAC and pRSTOP4Nsi_rev: 5′-ATAGGCATTTGTAGCAATAGAAAAAACGAACATAGATGCAAG.

Oligonucleotide pools (Genscript) were amplified using primers (Sigma): oligo_fwd: 5′-CTATTGCTACAAATGCCTATGCAGCCTCTTCATCTGGC and oligo_rev2: 5′-TCGTCACCCTCGGCGCCTCCTCCGGATCCTCCACC. Oligonucleotide PCR products were cloned into plasmid prSTOP4Nsi using NEB Builder HiFi Assembly (E5520, NEB) following the instructions of the manufacturer. The vector pool was used to transform electrocompetent ElectroMAX™ DH5α-E™ cells (11319019, Invitrogen). Plasmid DNA was harvested by MaxiPrep (GeneJet Plasmid Maxiprep Kit (K0491, Thermo Scientific)) and was transformed into SS320 phage display electrocompetent cells (60512, Lucigen), following the manufacturer's protocol, resuspended in 950 μL recovery media for each electroporation, transferred to 50 mL Falcon tubes, 100 μL $10^{11}$ CFU/mL M13KO7 helper phage (N0315S, NEB) added and shaken at 37 °C, 220 RPM for 1 h. 1 mL recovery culture was

added to 2YT medium containing 50 ug/mL carbenicillin (C9231, Sigma) and 25ug/mL kanamycin (K1377, Sigma), and grown overnight at 37 °C, 220 RPM. Phage was generated as described[70]. Briefly, phage was precipitated from the culture supernatant by adding 22.5 mL of supernatant to 4.5 mL 20% PEG 8000 (89510, Sigma)/2.5 M NaCl (S9625, Sigma), incubating on ice for 30 min, centrifuging at 12,857*g*, 4 °C for 10 min, resuspending the phage pellet in 1 mL of PBT (PBS (P4417, Sigma), BSA (A7030, Sigma) 0.20%, Tween 20 (P1379, Sigma) 0.05%), followed by centrifugation at 15,871*g*, 4 °C, for 10 min. Glycerol (G5516, Sigma) (10% final (v/v)) was added to supernatant, which was aliquoted and stored at −80 °C. For mutant libraries, 81-mer mutant oligonucleotides replacing NNK at each of the 16 peptide encoding residues were designed with the sequence arms described above. Oligonucleotides were individually amplified and cloned into plasmid prSTOP4Nsi.

## Phage display library screening
Phage display screening was performed using a protocol as described[71] with certain modifications. Experiments were performed in 96-well plates (Greiner Bio-One, 655180). Briefly, biotinylated chemokines (1 μg, Almac or ProteinFoundry, Supplementary Table 1) were immobilised on 5 μl streptavidin-coated magnetic beads (Dynabeads™ M-280 Streptavidin, Invitrogen, 11205D) and incubated overnight at 4 °C. Chemokine-bound beads were blocked for 2 h in blocking buffer (PBS (phosphate-buffered saline, Sigma, P4417) + 0.2% BSA (bovine serum albumin, Sigma, A7030), and phage library (100 μL) allowed to bind for 2 h at a concentration of $10^{10}$ CFU/mL. Beads were washed 15

times with PT buffer (PBS + 0.05% Tween 20 (Sigma, P1379)) to remove unbound phage and transferred to a fresh plate. Beads were incubated with 100 µL of phage-resistant Omnimax *E.coli* (Invitrogen, A10469) at optical density ($OD_{600}$) 0.6-0.8 and shaken at 37 °C for 30 min, following which 10 µL M13K07 helper phage (NEB, N0315S, final concentration $10^{10}$ cfu/mL) was added and shaken at 37 °C for 45 min. Cells were transferred to 1 mL of 2YT medium pH 7 (MP Biomedicals™, 113012031) supplemented with 150 µg/mL carbenicillin (Sigma, C9231) and 75 µg/mL kanamycin (Sigma, K1377) in a 96-deep well block with V-bottom (Corning, CLS3960) and grown overnight shaking at 37 °C, 200 RPM. Plates were either centrifuged at 2000*g* (4 °C for 30 min) or transferred to screw cap tubes and pelleted by centrifugation at 4000*g* and 4 °C for 15 min. 540 µL of supernatant was transferred to a new 96-deep well plate, 60 µL 10×PBT (10×PBS + 2% BSA + 0.5% Tween 20) added and stored at 4 °C until further use in subsequent rounds. Phage display was carried out for three rounds.

## Next generation sequencing
Inserts from the input library and from the final phage population following selection were amplified by PCR using primers: 5′-ACACTCTTTCCCTACACGACGCTCTTCCGATCTCTAGCGCTATGCCTATGCAGCCTCTTCA and 5′-GACTGGAGTTCAGACGTGTGCTCTTCCGATCTCGTCTGCGATGACAACAACCATCGCCCA, (Life Technologies). The PCR products were cleaned up using Monarch® PCR & DNA Cleanup Kit (5 µg) (T1030, NEB), 260/280 ratio determined by Nanodrop, quantified using a Qubit dsDNA Quantitation, High Sensitivity Kit (Q32851, Thermo Scientific) and subsequently sequenced at Azenta/GeneWiz using the AmpliconEZ protocol.

## Next generation sequence analysis
Paired end read Fastq files were joined by the read id. Sequences containing both forward and reverse demultiplexing regions (5′-CTAGCGCT and 5′-CGCAGACG) and both constant regions (5′-ATGCCTATGCAGCCTCTTCATCTGGC and 5′-GGTGGAGGATCCGGAGGAGGCGCCGAGGGTGACGATCCCGCAAAAGCGGCCTTTAACTCCCTGCAAGCCTCAGCGACCGAATATATCGGTTATGCGTGGGCGATGGTTGTTGTCAT were analysed. Insert sequences from each read were identified and reads with non-identical insert sequences or ambiguous sequences were excluded, as were inserts that were incorrect in size (i.e., not 48 nucleotides). Insert sequences were translated to peptide. Only inserts present in the designed library were counted and frequency determined. Library cloning efficiency was calculated as the ratio of distinct cloned peptide inserts to the number of inserts in the designed library and was 98%. Hexadecapeptide enrichment (E) following selection was calculated as ratio of output peptide frequency to input peptide frequency and expressed as log2E. Where proportion of count in input library was not available it was replaced with the lowest input proportion in the experiment rather than replacing it with 0. This is needed to avoid a value of E that is infinity. For mutagenesis experiments, where proportion of count in output was not available it was replaced with the lowest output proportion in the experiment rather than zero, which would result in a -Infinity log2E.

## Peptide mapping to parental protein sequences
Enriched peptides were mapped to parental proteins using custom R-scripts. Parental protein sequences were obtained from UniProt (https://www.uniprot.org) using RCurl. Briefly, collated enrichment data were filtered to include only those peptide-chemokine combinations where there was enrichment (i.e., log2E > 0), and wild-type and mutant peptides were mapped to the originating wild-type protein. Residue log2E was calculated as the sum of log2Es of peptides overlapping a residue. The rolling median of residue log2E was calculated using the "rollmedian" function in R-package zoo, with *k* = 7 and fill=NA. Confidence interval of the median was calculated using the "cimed" function in R-package asbio.

## Annotation of parental proteins
PDB files for evasin complexes with chemokines 7S4N and 7SO0 (for EV974) and 3FPU (for EVA1) were retrieved from RCSB PDB (https://www.rcsb.org) using bio3d[72] and chemokine-binding site residues were identified using the bio3d "binding.site" function with the default cut-off of 5 Å. AlphaFold models were downloaded from https://alphafold.ebi.ac.uk/download and structural features extracted using bio3d.

## Peptide sequence analysis
Tile-plots were constructed using pheatmap. Neighbour-joining trees were constructed by aligning peptide sequences using "ClustalW" with the Gonnet substitution matrix using R package msa, and aligned sequences were used to construct a matrix of pairwise distances. Neighbour-joining trees were constructed with 100 bootstrap replicates and midpoint-rooting using R packages ape, phytools, and ggtree. Peptide logos were constructed using R-packages ggmsa and ggseqlogo, and coloured using the Taylor colouring scheme.

## Peptides
Peptides for all cell migration experiments were obtained from GenScript at >95% purity and were synthesised using Fmoc solid-phase synthesis to give peptides with a C-terminal amide. LC-MS data provided by the supplier show that HD2 peptide is at the expected molecular weight i.e., it is monomeric (Supplementary Fig. 10). Peptide sequences are provided in Supplementary Table 2. The HD2SCR sequence TLETDTFYECPDAYAY was designed by generating 50 random shuffles using the function "stri_rand_shuffle" (R-package stringi_1.7.6) and then selecting the peptide with the maximal "osa" string distance to wild-type using R-package stringdist_0.9.8.

## Protein expression and purification
HIS:SUMO:peptide plasmids were transformed into BL21(DE3) cells (C2527H, NEB) and grown in 5 mL LB (L3022, Sigma) + 50 ug/mL Kanamycin (420411, Millipore) media overnight at 37 °C, with shaking at 200 RPM. 1% of this primary culture was inoculated into a secondary culture and grown for approximately 2 h at 37 °C, with shaking at 200 RPM. At OD600 of 0.4, the culture was induced with 1 mM IPTG (R0392, Thermo Scientific) and grown for 4 h at 37 °C, with shaking at 200 RPM. The induced cells were harvested at 3800 g in a Thermo Scientific Megafuge 16R using a swinging bucket rotor and the media was discarded. The pellet from a 200 mL culture was resuspended in 30 mL of binding buffer [Phosphate Buffer Saline, PBS (P4417, Sigma), 500 mM NaCl (S3014, Sigma), pH 7.2] containing 1 mM PMSF (P7626, Sigma). The suspension was sonicated using a Bandelin Sonopuls HD2070 for 45 min on ice using 0.7 s on- 0.3 s off cycle at 40% power. The lysate was centrifuged at 8500*g* in a Sorvall LYNX 4000 centrifuge for 20 min at 4 °C and the supernatant collected. 5 µL of DNase I (M0303, NEB) was added to the supernatant and incubated for 15 min in ice. Supernatant was filtered and passed over IMAC Sepharose 6 Fast Flow column resin (17-0921-07, GE Healthcare) [pre-treated with 0.2 M $NiCl_2$ (339350, Sigma) and equilibrated with binding buffer (PBS, 500 mM NaCl pH 7.2)]. The resin was incubated with the lysate supernatant for 1 h at 4 °C on a platform rocker for continuous mixing. 30 mL wash buffer (PBS, 500 mM NaCl, 20 mM imidazole (I5513, Sigma), pH 7.2) was applied to the resin. The peptide was then eluted in 5 mL elution buffer (PBS, 500 mM NaCl, 500 mM imidazole, pH 7.2), and buffer exchange in PBS was performed using Amicon Ultra Centrifugal Units 3 K (UFC9003, Millipore). EVA4 was expressed in HEK293F cells (R79007, ThermoFisher Scientific) by transient transfection of P1922 plasmid using polyethylenimine (24765, Polysciences) in HEK293F cells followed by culturing in Freestyle 293 expression medium (12338026, ThermoFisher) at 37 °C, 8% $CO_2$, at 130 rpm for 5 days. Protein was purified from filtered supernatants using nickel-charged IMAC Sepharose 6 Fast Flow resin as described above.

Elutions were concentrated using an Amicon Ultra-15 Centrifugal Filter Unit and purified by size exclusion chromatography (SEC) on an AKTA Start system using HiLoad 16/600 Superdex 75 (GE28-9893-33, GE Healthcare), in SEC buffer (PBS + 150 mM NaCl). Fractions showing absorption at 280 nm were analysed by electrophoresis on a Bolt Bis-Tris Plus Mini Protein Gel, 12% (NW00122, Invitrogen) in Bolt MES SDS Buffer (B000202, Invitrogen), stained with Quick Blue Protein Stain (LU001000, LuBio) and were pooled.

## Biolayer interferometry

The binding of His-SUMO tagged peptide fusions to various chemokines (Supplementary Table 3) was investigated by biolayer interferometry (BLI) using Ni-NTA biosensors (18-5101, ForteBio). All BLI data were obtained at 25 °C using a ForteBio-Sartorius Octet RED 384 instrument and Octet 384-well tilted bottom microplate (18-5080, Sartorius). The biosensors were preincubated overnight at room temperature in BLI buffer (PBS, 500 mM NaCl, 0.01% BSA (A7638, Sigma) + 0.002% Tween (P2287, Sigma)). His-SUMO tagged peptide fusions at 1 mg/mL (for cross-binding screen assay) and at 0.25 mg/mL (for kinetic assays) were immobilised on the Ni-NTA biosensors for 500 s using the BLI buffer. The HIS:SUMO fusion-loaded biosensors were then washed in the BLI buffer to allow signal stabilisation. To study the association with the analyte (chemokines), the biosensors were then dipped in chemokine solutions of various concentrations (1 µM for cross-binding screen assays, a range of 25–750 nM for kinetic assays) made in the BLI buffer, for either 300 s for cross-binding screen assays, or 600 s for kinetic assays. This was followed by the dissociation step, where the biosensors were incubated in BLI buffer for 300 s for screening assays and 600 s for kinetic assays. For chemokine cross-binding screens, wavelength shift was normalised by subtracting the signal from buffer control, and normalising the highest Rmax (i.e., with CCL8 and HD2) obtained in the experiment to 1. Kinetic data was analysed using buffer subtracted wavelength shift and a 1:1 binding model. We calculated mean $K_D$ and its standard error from at least three fits where full $R^2$ (i.e., how well the fit and experimental data correlate) was >0.95, full $X^2$ (i.e., measure of error between experimental data and fitted line) was <3, and $K_D$ standard error was <$K_D$.

## THP1 cell migration

For THP1 migration assays, 300000 cells/well were added to the top chamber of a 5-µm 96-well Transwell insert (3387, Corning) in 50 µL of cell migration media (RPMI-1640 (R0883, Sigma), 0.5% FBS (F9665, Sigma), 4 mM L-Glutamine (G7513, Sigma), 0.05% DMSO (D4540, Sigma)). The bottom chamber contained 150 µL of migration media with chemokine and peptide added. Cells were migrated at 37 °C in 5% $CO_2$ for 4 h.

## J:CXCR1 cell migration

J:CXCR1 migration assays were performed exactly as for THP1 assays except that a 3-µm Transwell insert (3385, Corning) was used.

## Activated T-cell migration

For activated-T cell migration assays, 100000 cells/well were added to the top chamber of a 3-µm Transwell insert (3385, Corning) in 50 µL of migration media (HBSS (14025-092, Life Technologies), 0.1% protease free bovine serum albumin (A7030, Sigma), 0.05% DMSO (D4540, Sigma)). The bottom chamber contained 150 µL of migration media with chemokine and peptide added. Cells were migrated at 37 °C in 5% $CO_2$ for 2 h.

## Analysis of cell migration

Following the migration assay the migration plate was shaken at 850 RPM for 10 min, and media from bottom plate transferred to a round-bottomed 96 well plate (353910, Falcon). Cell counts were determined using an Atune NxT Flow Cytometer Plate Reader with Cytkick

autosampler (ThermoFisher), based on cell size parameters FSC-H/SSC-H, and and a previously defined gate setting for each cell type (Supplementary Fig. 11). All experiments were performed as three technical and three biological replicates. Statistical significances between control and experimental groups was evaluated using Dunnett's test[73]. $IC_{50}$ experiments were performed at the $EC_{80}$ dose of chemokine. $EC_{80}$ was calculated by fitting a chemokine dose-response curve with a 3-parameter log-logistic model (fixing the top to 100%) using the function "drm" in R-package drc[74]. $IC_{50}$ was calculated by fitting an inhibitor response curve with a 4-parameter log-logistic model using the function "drm" in R-package drc[74]. All $IC_{50}$ values reported had $p$-value < 0.05.

## Modelling of chemokine complexes with peptides, receptors and evasins

Models were generated using AlphaFold2-Multimer at the COSMIC[2] Science Gateway using default parameters and a full database search using UniProt mature protein sequences[50,75]. The five highest confidence models were identified in each case from the confidence score and carried forward for heat map generation. As experimental structures often contain missing sidechain atoms, direct use of these structures will interfere with the atom-wise scoring function of Auto-Dock CrankPep (ADCP), and, also, as certain chemokines e.g., CXCL6, lack PDB structures, to maintain consistency, we first generated chemokine structures using AlphaFold for input to ADCP. The highest-ranking structure for each chemokine was reduced with "reduce", then the functions "prepare_receptor", "agfr", and "adcp" were called from ADFRsuite 1.0. Default parameters were used for "prepare_receptor" and "agfr", while "adcp" was run with $N = 300$, $n = 48000000$ and nc = 0.8 for all runs. For ADCP the ten highest ranked poses (based on automatically calculated lowest free energy) were carried forward for heat map generation. PDB files from previously reported HADDOCK[76] analysis of NMR chemical shift perturbation data of peptide Ev4Glu[14]-Asn[31] in complex with CCL5[28] were kindly provided by Prof. Ingrid Dijkgraaf, Maastricht. Binding site heatmaps were generated by identifying residues within 5 Å distance of the two chains using bio3d[72] for each docking pose. Weighted proximity scores were calculated as follows: A per-residue-score equal to the confidence score (AlphaFold) or calculated free energy (ADCP) was ascribed for each pose within a model, this was aggregated over the different poses in the model, and then normalised to a maximum score of 100 to allow comparison between models. For NMR-HADDOCK models the 4 generated poses were weighted equally for the calculation of the weighted proximity score. Docked poses were aligned using PyMol "extra_fit" to align multiple poses to the top-ranked pose.

## Arpeggio analysis

Arpeggio[64] contacts between the two chains of the five AlphaFold multimer models for each complex were identified from the ".bs_contacts" file produced. Column identities for the file were taken from https://bitbucket.org/harryjubb/arpeggio/src/master/README.md.

## Rosetta analysis

We performed model relaxation using Rosetta 3.13 (relax.static.linuxgccrelease) followed by calculation of the cross-interface binding energy (dG_cross) with Rosetta InterfaceAnalyser (InterfaceAnalyzer.static.linuxgccrelease) using the default ref2015 weightset[77–79].

## Statistical information

For cell-based experiments sample size was not formally calculated. Cell numbers, numbers of technical and biological replicates are based on optimisation of such experiments[20–23,34]. Statistical analyses were performed using the R-base package stats and DescTools. Statistical tests used were one-way ANOVA followed by a two-sided

Dunnett's post-hoc multiple comparison procedure for comparing several treatments with a control, with 95% family wise confidence level[73]. Data are displayed as Tukey box-whisker plots using the ggplot function geom_boxplot, and displays median, lower and upper hinges (25th and 75th percentiles), and whiskers from hinge to 1.5* interquartile range.

## Software

Biolayer interferometry data was collected and analysed using ForteBio Data Analysis HT 11.1. Flow sorting data was collected and analysed using Attune Cytometric Software v5.1.1. Structural models were generated using AlphaFold2-Multimer at the COSMIC[2] Science Gateway[50,75], and ADFRSuite 1.0[51] (https://ccsb.scripps.edu/adcp/downloads/) and reduce ((https://github.com/rlabduke/reduce) on the Oxford University BioMedical Research Computing Cluster (BMRCC). Data analysis was performed using R version 4.2.3, RStudio version 2023.03.2 + 454, running on arch64-apple-darwin20 (64-bit) with macOS Ventura 13.4.1. R packages used were: ape_5.7-1, asbio_1.9-2, bio3d_2.4-4, Biobase_2.58.0, BiocGenerics_0.44.0, BiocParallel_1.32.6, Biostrings_2.66.0, dendsort_0.3.4, DescTools_0.99.48, dplyr_1.1.2, drc_3.0-1, flextable_0.9.1, forcats_1.0.0, GenomeInfoDb_1.34.9, GenomicAlignments_1.34.1, GenomicRanges_1.50.2, ggmsa_1.4.0, ggnewscale_0.4.8, ggplot2_3.4.2, ggpubr_0.6.0, ggseqlogo_0.1, ggtree_3.6.2, ggupset_0.3.0, gridExtra_2.3, gtable_0.3.3, IRanges_2.32.0, janitor_2.2.0, jsonlite_1.8.4, lubridate_1.9.2, maps_3.4.1, MASS_7.3-58.2, MatrixGenerics_1.10.0, matrixStats_0.63.0, msa_1.30.1, officer_0.6.2, openxlsx_4.2.5.2, pals_1.7, pheatmap_1.0.12, phytools_1.5-1, purrr_1.0.1, R.methodsS3_1.8.2, R.oo_1.25.0, R.utils_2.12.2, RColorBrewer_1.1-3, readr_2.1.4, Rsamtools_2.14.0, S4Vectors_0.36.2, scales_1.2.1, seqinr_4.2-30, ShortRead_1.56.1, strex_1.6.0, stringr_1.5.0, SummarizedExperiment_1.28.0, tibble_3.2.1, tidyr_1.3.0, tidyverse_2.0.0, viridis_0.6.3, viridisLite_0.4.1, XVector_0.38.0, zoo_1.8-12. Arpeggio analysis used Python 2.7, biopython 1.79, and Arpeggio[64] (https://bitbucket.org/harryjubb/arpeggio/src/master/). Open-Source PyMOL (https://pymolwiki.org/index.php/MAC_Install) was used for scripting and PyMOL 2.5.2 (https://pymol.org/2/) was used for visualization of structural models. Rosetta analysis used Rosetta 3.13[77].

## Reporting summary

Further information on research design is available in the Nature Portfolio Reporting Summary linked to this article.

# Data availability

The authors declare that the data supporting the findings of this study are available within the paper and its supplementary files. Plasmids described and sequences are available on request from the corresponding author. Previously published structures used in our analyses were obtained from the PDB with accession codes as follows: 3FPU, 7S4N, 7SO0. Source data are provided with this paper.

# Code availability

Code used in this study is available in the source data provided with this paper.

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

## Acknowledgements

This research was funded by British Heart Foundation Chair Award (CH/09/003/26631), BHF Program Grant (RG/18/1/33351), Oxford BHF Regenerative Medicine Award (RM/17/2/33380) and Oxford BHF Centre of Research Excellence (RE/13/1/30181) Awards to S.B. M.P. is funded by a studentship from the Radcliffe Department of Medicine and the Clarendon Scholarship Fund and CC by a studentship from the Wellcome Programme in Genomic Medicine and Statistics. S.D. is funded by a Rubicon Fellowship from NWO. C.O. is supported by the Engineering and Physical Sciences Research Council through a postdoctoral fellowship (EP/W522582/1), and by Schmidt Futures through an Eric and Wendy Schmidt AI in Science Postdoctoral Fellowship. The research on the BMRC Cluster was supported by the Wellcome Trust Core Award Grant Number 203141/Z/16/Z with additional support from the NIHR Oxford BRC. The views expressed are those of the author(s) and not necessarily those of the NHS, the NIHR or the Department of Health. The authors are grateful to Prof. Ingrid Dijkgraaf (Maastricht) for kindly making available PDB files from previously published[28] NMR/HADDOCK analysis.

## Author contributions

Phage-display and mutant libraries were designed by S.B. and constructed by G.D., K.H. and J.K. Phage-display screens were performed by K.H., S.V. and J.K. NextGen sequencing data analysis was performed by S.B. BLI was performed by S.C. The cell lines J:CXCR1 and ATC were generated by G.S. and M.P., respectively. Chemotaxis assays were performed by S.V., M.P., J.K., P.M. and G.S. S.D. synthesised peptides. S.B. performed bioinformatic and statistical analyses. Computational modelling was performed by C.C., S.D. and S.B. with advice from G.M., C.O., and C.D. on establishing the modelling pipeline. S.B., G.D., and G.M. supervised and interpreted experiments and modelling outputs. All authors contributed to writing the manuscript.

## Competing interests

The authors declare no competing interests.
