## [Peer Review File · Nature Communications]

REVIEWER COMMENTS

Reviewer #1 (Remarks to the Author):

Review manuscript Nature Communications

Titel: Discovery and pharmacophoric characterization of chemokine network inhibitors using phage display, saturation mutagenesis and computational modelling

Authors: Serena Vales et al.

In this study, the authors developed a pipeline combining phage-display, saturation mutagenesis and computational modelling to discover promiscuous chemokine-binding peptides from class A evasins and characterize the pharmacophore.

Overall this is a relevant study that revealed novel peptides that bind to chemokines from different chemokine classes. The interaction sites of chemokines and developed peptides have been analyzed thoroughly. The methods used answer the research questions and the results support the conclusions.

The manuscript is written very clearly.

Comments:

There are both inflammatory and homeostatic chemokines. This should be mentioned in the manuscript (discussion section?).

How did the authors designed the HD2SCR scrambled peptide? By a crambled peptide sequence generator?

Why did the authors start from class A and not from class B evasins? This is not mentioned.

Do the authors have an explanation for the different binding patterns of the HD-peptides to SDF1 and its isoform SDF1B?

Is it necessary to use hexadecapeptides or are in future shorter (or longer) peptides foreseen? And would a cyclic hexadecapeptide be an option? For clinical use, a peptidic compound should be resistant towards proteolytic degradation.

The unpaired Cys residue is apparently important for binding. The authors have investigated this by changing the Cys by an Ala and Ser, which caused loss of affinity. However, unpaired Cys residues usually give problems like formation of dimers etc. Have the authors observed formation of dimers? Now, the authors used Ala and Cys replacement as a negative control. The importance of Cys could be confirmed by the use of selenocysteine, because selenocysteine will interact more rapidly than cysteine.

To further elucidate the pharmacophore, the biochemical studies were complemented with two different *in silico* approaches. The impact of the study would improve by confirmation of interaction sites and SAR with NMR or X-ray crystallography.

Reviewer #2 (Remarks to the Author):

Review for Vales et al. NCOMM

In this paper, the authors use phage display to screen a library of peptide fragments against 25 different chemokines to identify some peptides that appear to bind many (over 15) different chemokines from 2 families (CC and CXC). They confirm binding and inhibition and identify positions important for binding in one promiscuous candidate. They finally obtain computational models of this candidate binding to different chemokines.

I cannot much comment on the significance of the findings in the chemokine field (due to lack of background); from a technical point of view, I find this a pretty thorough study (with some reservations, see below), albeit without much methodological innovation. As an application of existing methods to a new field, I would be in favour of publication if biological significance of the results is given (I suspect yes, but don't really know, see above).

A few points that the authors could further address:

- 1) The authors have mutational scanning data on HD2 and could've checked whether their docking generated models are consistent with their mutational scanning data - the easiest would be to

recapitulate the mutants seen in the scanning data as highly affecting / not affecting binding computationally and check their effects.

2) The BLI affinity measurements are a bit unsatisfying; the authors should report proper K_d values for all the chemokines that they want to claim HD2 binds to; from Fig 4 it is difficult to conclude that they observe binding for any protein other than CCL8.

3) It would be helpful for the authors to establish that their log₂E enrichment measurement actually correlates with a biophysical characteristic of binding (presumably K_d).

4) For their docking models, it would be good to ensure that they are consistent with the models that, e.g., AF2-multimer would produce for EVA4 (and in turn, whether these models would be consistent with previous information, e.g., in ref #32).

Reviewer #3 (Remarks to the Author):

Vales et al. utilize a phage display screening approach based on peptides from class A tick evasins to identify peptides with promiscuous binding of chemokines. They find a number of peptides which bind a range of CC as well as CXC chemokines and show through mutagenesis and molecular modelling that the peptides bind chemokines from the different classes using distinct epitopes.

The manuscript is thorough, the amount of data extensive and well displayed. Application of the phage-display method to screen peptide binding to multiple chemokines and identification of peptides with enrichment scores correlating with chemokine affinity is noteworthy.

My major concerns are with significance of the findings. With CCL8 as an exception, the chemokine affinities of the peptides are rather low based on the BLI data as well as the IC₅₀ values in the migration assay, which reduces the potential that evasin-derived peptides have therapeutic potential. The “example” peptide HA2 is similar to previously characterized peptides. Therefore, the most significant contribution of the manuscript is the method rather than the identified peptides. In addition, I have concerns with the interpretation of some of the results as described in my specific comments below.

Specific comments:

One major finding of the manuscript is that peptides derived from class A evasins can bind CXC chemokines but the parental evasins cannot. Is this true at the high concentrations where the peptide inhibits migration (no inhibition of CXCL10-induced migration below 2 μ M in Fig 6D)? The authors state

on line 175 that EVA4 did not inhibit CXCL10 and therefore no IC50 was determined. This data is not shown and the range of the EVA4 concentrations tested is not mentioned in the methods. The authors need to show that EVA4 does not inhibit CXCL10-driven migration at uM concentrations to support the conclusion that HD2 binds and inhibits CXCL10 but EVA4 does not.

The authors primarily characterize HA2, which is in fact nearly identical (two residues shorter and no C to A mutation) to the EVA4 Glu14–Asn31 peptide characterized by Denisov et al (ref 32) for binding to CCL5. I think this should be acknowledged and the results Glu14–Asn31 discussed in the context of their models.

The nomenclature of CXC chemokines is inconsistent, for example, CXCL10 is sometimes referred to as CXL10 and sometimes as CXCL10. In addition, in contrast to the other chemokines, SDF and IL8 are used instead of CXCL12 and CXCL8. This should be adjusted and consistent in the revised manuscript with the CXC, CC nomenclature (e.g CXCL10) used for all chemokines.

Line 152: I don't understand how the authors conclude that the Cys residue contributes to promiscuity rather than likely being important for binding affinity

Line 142: The authors note that there is a difference between SDFalpha and beta in their screen. This could be due to the tag placement affecting CXCL12alpha as has been seen before for binding of biotinylated CXCL12 to ACKR3 (Gustavsson et al. *Sci Signal*. 2019;12(598):eaaw3657) and S6-tagged CXCL12 to CXCR4 (Figure 2 in Kawamura et al. *PLoS One*. 2014 28;9(1):e81454). CXCL12beta has additional residues at the C-terminus and thus the tag is further removed from the core of the chemokine and the proposed evasin-binding epitope, which could reduce the effect of the tag.

The analysis of the chemokine regions in contact with the peptide is quite superficial. I think adding a sequence alignment for the different chemokines using the residues with the highest weighted proximity score would add important information and could also potentially answer, for example, why CCL8 is the chemokine with the least loss of affinity compared to full length evasins.

Line 246: The recent structure of ACKR3: CXCL12 (Yen et al. *Sci Adv*. 2022 Jul 15;8(28):eabn8063) showed that the model in ref 48 is incorrect in its placement of the chemokine and receptor N-terminus. A better reference for this statement is the CXCR4: CXCL12 model published by Ngo et al. (*PLoS Biol*. 2020 Apr 9;18(4):e3000656) where the wrapping of the receptor N-terminus around the chemokine was experimentally validated.

REVIEWER COMMENTS

Reviewer #1 (Remarks to the Author):

Review manuscript Nature Communications

Titel: Discovery and pharmacophoric characterization of chemokine network inhibitors using phage display, saturation mutagenesis and computational modelling

Authors: Serena Vales et al.

In this study, the authors developed a pipeline combining phage-display, saturation mutagenesis and computational modelling to discover promiscuous chemokine-binding peptides from class A evasins and characterize the pharmacophore.

Overall this is a relevant study that revealed novel peptides that bind to chemokines from different chemokine classes. The interaction sites of chemokines and developed peptides have been analyzed thoroughly. The methods used answer the research questions and the results support the conclusions.

The manuscript is written very clearly.

Comments:

There are both inflammatory and homeostatic chemokines. This should be mentioned in the manuscript (discussion section?).

We have now explained the differences between inflammatory and homeostatic chemokines in the introduction (lines 37 - 40) and have also indicated the type (inflammatory or homeostatic) in Fig. 3A as an annotation to the heatmap.

How did the authors designed the HD2SCR scrambled peptide? By a crambled peptide sequence generator?

We have now described the method (using "stri_rand_shuffle" and "osa" (R-packages) in the methods section (lines 509-511).

Why did the authors start from class A and not from class B evasins? This is not mentioned.

We started from class A evasins as linear chemokine-binding peptides had already been identified from this evasin class using structural methods (HDX-MS, NMR), and this would give us an opportunity to test the phage-display method. We have now explained the rationale in the results section (lines 119 - 121).

Do the authors have an explanation for the different binding patterns of the HD-peptides to SDF1 and its isoform SDF1B?

It is likely that tag placement for the different isoforms results in different binding patterns. We have now added an explanation with references into the revised manuscript (lines 151 - 153).

Is it necessary to use hexadecapeptides or are in future shorter (or longer) peptides foreseen? And would a cyclic hexadecapeptide be an option? For clinical use, a peptidic compound should be resistant towards proteolytic degradation.

Indeed, the pharmacophoric features identified could be exploited to develop shorter linear peptides, cyclic peptides and peptidomimetics – i.e. synthetic agents that mimic the 3D-spatial and physicochemical properties of parental peptides but overcome their pharmacokinetic limitations. We have added a sentence to the discussion to indicate this (lines 380-384).

The unpaired Cys residue is apparently important for binding. The authors have investigated this by changing the Cys by an Ala and Ser, which caused loss of affinity. However, unpaired Cys residues usually give problems like formation of dimers etc. Have the authors observed formation of dimers? Now, the authors used Ala and Cys replacement as a negative control. The importance of Cys could be confirmed by the use of selenocysteine, because selenocysteine will interact more rapidly than cysteine.

LC-MS data provided by the supplier show that HD2 peptide is at the expected molecular weight i.e. it is monomeric. We have added a sentence to the methods to clarify this (lines 505-507), and Supplementary Fig. 10. Following on from the suggestion to explore the substitution of HD2 Cys with selenocysteine, we attempted to synthesise this in-house, but unfortunately this was not successful. Repeated attempts at synthesis in two commercial facilities (Genscript and Biosynth) have also failed. We regret that we are thus unable to explore this interesting approach at the present time.

To further elucidate the pharmacophore, the biochemical studies were complemented with two different *in silico* approaches. The impact of the study would improve by confirmation of interaction sites and SAR with NMR or X-ray crystallography.

We agree that such experiments would improve the impact of the study. Such experiments require substantial NMR / X-ray-crystallography experimentation. To perform the NMR/ X-ray-crystallography satisfactorily would require NMR/XRay analysis of HD2 and key Ala mutations with at least several different CC and CXC-chemokines where inhibitory activity has been demonstrated. This is a major undertaking which will require substantial future work. We have indicated this as a limitation of the study in the discussion (lines 370-372).

Reviewer #2 (Remarks to the Author):

Review for Vales et al. NCOMM

In this paper, the authors use phage display to screen a library of peptide fragments against 25 different chemokines to identify some peptides that appear to bind many (over 15) different chemokine from 2 families (CC and CXC). They confirm binding and inhibition and identify positions important for binding in one promiscuous candidate. They finally obtain computational models of this candidate binding to different chemokine.

I cannot much comment on the significance of the findings in the chemokine field (due to lack of background); from a technical point of view, I find this a pretty thorough study (with some reservations, see below), albeit without much methodological innovation. As an application of existing methods to a new field, I would be in favour of publication if biological significance of the results is given (I suspect yes, but don't really know, see above).

We would like to take this opportunity to highlight that the application of phage-display NNK saturation mutagenesis and deep sequencing to perform alanine-, hydrophobic- and hydrophilic- scanning to systematically identify key peptide pharmacophoric features (e.g., anionic, hydrophobic, unpaired cysteine residues) represents a methodological advance as to our knowledge this has not been previously demonstrated systematically.

A few points that the authors could further address:

1) The authors have mutational scanning data on HD2 and could've checked whether their docking generated models are consistent with their mutational scanning data - the easiest would be to recapitulate the mutants seen in the scanning data as highly affecting / not affecting binding computationally and check their effects.

We agree with this suggestion. The state-of-art to determine binding affinity using computational approaches is molecular dynamics simulation¹. To obtain sufficient data points to correlate with the mutagenesis data would require MD computational analysis of HD2 and 14 Alanine-mutants against 24 chemokines, (360 dockings), typically done in triplicate. This is time-demanding and requires substantial computational resources and we regret that it is not practicable within the time-frame for revision, but is instead suitable for future work.

2) The BLI affinity measurements are a bit unsatisfying; the authors should report proper K_d values for all the chemokines that they want to claim HD2 binds to; from Fig 4 it is difficult to conclude that they observe binding for any protein other than CCL8.

We now provide an extended series of BLI affinity measurements in the revised version of the manuscript (Fig 4, Supporting Information Table 4) for those chemokines where we could reproducibly

detect binding in cross-binding screens (Supporting information Fig. 2). We have revised the relevant section of the manuscript (lines 165-192).

3) It would be helpful for the authors to establish that their log₂E enrichment measurement actually correlates with a biophysical characteristic of binding (presumably K_d).

The phage-display screening experiment (Figs. 1-3) is designed to identify peptides that enrich following selection with a *single* bait in comparison to the input phage library. As referenced in the manuscript supporting our use of log₂E (lines 129-131):

"Following library selection with each chemokine we calculated the enrichment (E) of each peptide in comparison to the input library, and expressed the enrichment as log₂E, as this metric is correlated with binding affinity ²."

However, this correlation refers to data obtained by a *single* bait, and to make comparisons *between* baits the data need to be normalized to an internal control. As these were screening experiments, they lacked an internal control to which log₂E could be normalised. Similarly, comparisons *between* methods are possible only when the same internal control is present in both experimental approaches allowing for appropriate normalisation of data to the internal control. Consequently, we regret that it is not possible to perform correlation between log₂E obtained in the phage-display screening experiment and K_d obtained from BLI.

We did perform correlation assays in experiments where internal controls were present. For instance, in the Ala-mutagenesis phage-display experiment (Fig 7, 8) the library included the wild-type peptide HD2 as an internal control. We were thus able to measure Δlog₂E - i.e., the change in log₂E of a mutation compared to the internal control for every chemokine tested. As shown, (Fig. 8c), Δlog₂E correlates strongly and significantly with the ability to inhibit chemotaxis as measured by Δmigrated cell count (the change in comparison to the internal control, i.e., wild-type HD2), in the chemotaxis assays.

4) For their docking models, it would be good to ensure that they are consistent with the models that, e.g., AF2-multimer would produce for EVA4 (and in turn, whether these models would be consistent with previous information, e.g., in ref #32).

We have provided new analyses of chemokine:EVA4 complexes using AlphaFold2-Multimer (for chemokines known to bind/inhibited by EVA4, Supplementary Fig. 9), and show that these are consistent with the corresponding chemokine:HD2 models, and also that these models are consistent with previous information. We have revised the manuscript section to indicate this (lines 286-291).

Reviewer #3 (Remarks to the Author):

Vales et al. utilize a phage display screening approach based on peptides from class A tick evasins to identify peptides with promiscuous binding of chemokines. They find a number of peptides which bind a range of CC as well as CXC chemokines and show through mutagenesis and molecular modelling that the peptides bind chemokines from the different classes using distinct epitopes.

The manuscript is thorough, the amount of data extensive and well displayed. Application of the phage-display method to screen peptide binding to multiple chemokines and identification of peptides with enrichment scores correlating with chemokine affinity is noteworthy.

My major concerns are with significance of the findings. With CCL8 as an exception, the chemokine affinities of the peptides are rather low based on the BLI data as well as the IC₅₀ values in the migration assay, which reduces the potential that evasin-derived peptides have therapeutic potential.

We believe that the significance of the findings is that unlike the parental evasin EVA4, the peptide HD2 can bind and inhibit both CC and CXC class chemokines. We accept that the affinity and inhibitory potency are indeed not yet in the range desirable for therapeutic agents. Using saturation mutagenesis (Fig. 7d, 7f and unpublished observations), we have identified mutations of HD2 (e.g., anionic mutations at T6, hydrophobic mutations at L11) that enhance binding and inhibitory potency. Such "improving" mutations and combinations of such mutations, together with Fc-conjugation to

create peptibodies, are thus much more likely to be therapeutically effective. We have modified the discussion to emphasise this point (lines 384 - 393).

The “example” peptide HA2 is similar to previously characterized peptides. Therefore, the most significant contribution of the manuscript is the method rather than the identified peptides.

The example peptide HD2 is from the same region of evasin 4 as the octadecapeptide described in ref 32. (Denisov JBC 2020). Unlike HD2, this octadecapeptide has a Cys-Ala mutation, and was only shown to inhibit a single chemokine, CCL5. As shown in our manuscript, the example peptide HD2 is also similar to the EV672-derived peptides HD540 and HD845. HD845 was previously described by us as BK1.5 (ref 31, Darlot JBC 202), and together with other peptides of the BK series, were designed based on HDX-MS analysis of the EV672:CCL8 interface. None of the previously described evasinA - derived peptides were shown to have anti-chemokine activity against CXC-class chemokines. Thus, in addition to the method, the dual activity against both CC and CXC class chemokines, and the pharmacophoric characterisation by saturation mutagenesis, are the significant advances presented in this study.

We have now explained in more depth the relationship of HD2 to previously described peptides and highlighted the unexpected dual-activity against CC and CXC-class chemokines in the revised discussion (lines 317-326).

In addition, I have concerns with the interpretation of some of the results as described in my specific comments below.

Specific comments:

One major finding of the manuscript is that peptides derived from class A evasins can bind CXC chemokines but the parental evasins cannot. Is this true at the high concentrations where the peptide inhibits migration (no inhibition of CXCL10-induced migration below 2uM in Fig 6D)? The authors state on line 175 that EVA4 did not inhibit CXCL10 and therefore no IC50 was determined. This data is not shown and the range of the EVA4 concentrations tested is not mentioned in the methods. The authors need to show that EVA4 does not inhibit CXCL10-driven migration at uM concentrations to support the conclusion that HD2 binds and inhibits CXCL10 but EVA4 does not.

We have now added the data to the revised version of Fig. 6, which shows that while EVA4 inhibits CC chemokines it does not inhibit CXCL10 or CXCL6, whereas HD2, at the same concentration, does. We have modified the manuscript results section accordingly (lines 202-204).

The authors primarily characterize HA2, which is in fact nearly identical (two residues shorter and no C to A mutation) to the EVA4 Glu14–Asn31 peptide characterized by Denisov et al (ref 32) for binding to CCL5. I think this should be acknowledged and the results Glu14–Asn31 discussed in the context of their models.

We have added the acknowledgement to the discussion section (lines 319-326).

The nomenclature of CXC chemokines is inconsistent, for example, CXCL10 is sometimes referred to as CXL10 and sometimes as CXCL10. In addition, in contrast to the other chemokines, SDF and IL8 are used instead of CXCL12 and CXCL8. This should be adjusted and consistent in the revised manuscript with the CXC, CC nomenclature (e.g CXCL10) used for all chemokines.

We have revised the manuscript and figures to conform to IUPHAR nomenclature for chemokines.

Line 152: I don't understand how the authors conclude that the Cys residue contributes to promiscuity rather than likely being important for binding affinity

We apologise for the error and have revised the statement accordingly (lines 163-164).

Line 142: The authors note that there is a difference between SDFalpha and beta in their screen. This could be due to the tag placement affecting CXCL12alpha as has been seen before for binding of biotinylated CXCL12 to ACKR3 (Gustavsson et al. Sci Signal. 2019;12(598):eaaw3657) and S6-

tagged CXCL12 to CXCR4 (Figure 2 in Kawamura et al. PLoS One. 2014 28;9(1):e81454). CXCL12beta has additional residues at the C-terminus and thus the tag is further removed from the core of the chemokine and the proposed evasin-binding epitope, which could reduce the effect of the tag.

We thank the reviewer for pointing this out and have added these comments as a likely explanation for the differences between CXCL12 (SDF1) and CXCL12B (SDF1B) (lines 151-156).

The analysis of the chemokine regions in contact with the peptide is quite superficial. I think adding a sequence alignment for the different chemokines using the residues with the highest weighted proximity score would add important information and could also potentially answer, for example, why CCL8 is the chemokine with the least loss of affinity compared to full length evasins.

We have added a new figure (Fig. 9c) which comprises a chemokine sequence alignment together with the weighted proximity scores for peptide and for receptors. This clearly shows how peptide-proximal and receptor-proximal regions of the chemokines overlap for several different chemokines, indicating that the mechanism of action of HD2 is by steric hindrance. We have revised the manuscript results section accordingly (lines 301-303).

Line 246: The recent structure of ACKR3: CXCL12 (Yen et al. Sci Adv. 2022 Jul 15;8(28):eabn8063) showed that the model in ref 48 is incorrect in its placement of the chemokine and receptor N-terminus. A better reference for this statement is the CXCR4: CXCL12 model published by Ngo et al. (PLoS Biol. 2020 Apr 9;18(4):e3000656) where the wrapping of the receptor N-terminus around the chemokine was experimentally validated.

We thank the reviewer for pointing this out and have replaced reference 48 with the reference to Ngo et al. (PLoS Biol. 2020 Apr 9;18(4):e3000656). We have revised the manuscript results section accordingly (lines 298-301).

1. Jespers, W., Aqvist, J. & Gutierrez-de-Teran, H. Free Energy Calculations for Protein-Ligand Binding Prediction. *Methods Mol Biol* **2266**, 203-226 (2021).
2. Rogers, J.M., Passioura, T. & Suga, H. Nonproteinogenic deep mutational scanning of linear and cyclic peptides. *Proceedings of the National Academy of Sciences of the United States of America* **115**, 10959-10964 (2018).
3. Gustavsson, M., Dyer, D.P., Zhao, C. & Handel, T.M. Kinetics of CXCL12 binding to atypical chemokine receptor 3 reveal a role for the receptor N terminus in chemokine binding. *Sci Signal* **12** (2019).
4. Kawamura, T. et al. A general method for site specific fluorescent labeling of recombinant chemokines. *PLoS One* **9**, e81454 (2014).

REVIEWERS' COMMENTS

Reviewer #1 (Remarks to the Author):

The comments of this reviewer have been well addressed by the authors.

Reviewer #2 (Remarks to the Author):

Review for Vales et al. NCOMM

While the authors addressed many of my comments, I would respectfully disagree with the assertion that "phage-display ... to systematically identify key peptide pharmacophoric features ... represents a methodological advance". In my view large combinatorial display library screens (most often phage) have been carried out to find peptides or proteins binders in a very large variety of guises for decades now. This doesn't diminish the fact that is a valuable and powerful technique.

I also find it a bit disappointing that when I asked for consistency checks of their docking models, the authors seem to think that "not practicable". Even a dynamics-based (I'm assuming the authors mean FEP/TI) would be feasible, but much simpler methods (like Rosetta) would fine and would give much added confidence in their docking models. Without it, the docks really don't add much value.

Otherwise, my comments have been addressed. I would concur with reviewer #3 that affinities and potencies are pretty low, but other than that and the points above this appears to be a thorough study.

Reviewer #3 (Remarks to the Author):

With the revised manuscript and rebuttal letter the authors have answered my concerns about the manuscript and I support it being published.

I have one concern on the analysis of some of the data introduced in the revised manuscript. My comment is rather long but I wanted to make sure to explain my reasoning:

The revised manuscript has fits of the BLI data with one or two Kd values depending on if a 2:1 or 1:1 model was used to fit the data. This analysis is a nice qualitative and, in the case of CCL8, quantitative, inclusion in the manuscript. However, I am not convinced by the reported fits and the respective Kd values for the other chemokines. For example, CCL5 binding is increased around three-fold when increasing the concentration from 100 to 500nM, which is not consistent with the fitted Kds of 54 and 106nM. This can be compared to CCL8 where binding is increased by around 50% when CCL8 concentration is increased from 100 to 500nM, which indicates a higher affinity for CCL8 than CCL5. However, according to the fits CCL5 has higher affinity, which does not pass the eye test.

Multiple factors could affect this. For example, oligomerization of CCL5 will be increased at the higher concentration and dimer or higher oligomer binding, which has been previously observed for CCL5, would give a larger signal response. In general, the curves at high chemokine concentration appears to have slower kobs with signal accumulating over time than those at lower chemokine concentration, which is a sign that there are oligomer or other potentially non-specific contributions at high chemokine concentrations. For an interaction following pseudo-first order kinetics kobs would expected to increase with increasing chemokine concentration, as seen for CCL8.

In summary, the BLI data convincingly shows that the chemokines bind the peptide but (except for CCL8) the use of a 2:1 binding model in the analysis and the reporting of multiple exact Kd values is, in my opinion, not appropriate based on the presented data.

Comments and responses to revised manuscript

Reviewer #1 (Remarks to the Author):

The comments of this reviewer have been well addressed by the authors.

Reviewer #2 (Remarks to the Author):

Review for Vales et al. NCOMM

While the authors addressed many of my comments, I would respectfully disagree with the assertion that "phage-display ... to systematically identify key peptide pharmacophoric features ... represents a methodological advance". In my view large combinatorial display library screens (most often phage) have been carried out to find peptides or proteins binders in a very large variety of guises for decades now. This doesn't diminish the fact that is a valuable and powerful technique.

Response: We agree that phage display screens have been performed extensively in the past. However, we have not found evidence in the published literature that the specific method we describe, i.e., "phage-display NNK saturation mutagenesis and deep sequencing to perform alanine-, hydrophobic- and hydrophilic- scanning", has been previously used to systematically identify key peptide pharmacophoric features. To avoid contention, we have not made any assertion regarding this in the manuscript itself.

I also find it a bit disappointing that when I asked for consistency checks of their docking models, the authors seem to think that "not practicable". Even a dynamics-based (I'm assuming the authors mean FEP/TI) would be feasible, but much simpler methods (like Rosetta) would fine and would give much added confidence in their docking models. Without it, the docks really don't add much value.

Response: As suggested we have performed a Rosetta analysis of the docked models, the analysis is provided in Supplementary information Table 7. We have added the following lines [267-271] to the manuscript text: "We calculated the Rosetta cross-interface binding energy for each docked structure (Supplementary information Table 7) as this parameter shows highest AlphaFold model classification accuracy¹. In all cases but one the cross-interface binding energy was less than -16, the suggested cut-off value¹, supporting the docked models. "

We have confidence in the docking models as:

- a)** We employed two paradigmatically distinct methods - AlphaFold2-Multimer (using deep learning) and AutoDock CrankPep (which folds flexible peptides and docks them into rigid targets) - to come to our conclusions;
- b)** The results of our two docking models with CCL5 are highly consistent with the NMR-derived model of EvGlu14-Asn31 (which differs from HD2 by 3 residues) in complex with CCL5 reported previously in Denisov et al ². This is made clearer in the revised version of the manuscript (Figure 10).

Furthermore, we respectfully argue that these docks still add value independently from scoring, as they provide insight into the sterically accessible conformations that a peptide binder may take relative to the chemokine.

Reviewer #3 (Remarks to the Author):

With the revised manuscript and rebuttal letter the authors have answered my concerns about the manuscript and I support it being published.

I have one concern on the analysis of some of the data introduced in the revised manuscript. My comment is rather long but I wanted to make sure to explain my reasoning:

The revised manuscript has fits of the BLI data with one or two Kd values depending on if a 2:1 or 1:1 model was used to fit the data. This analysis is a nice qualitative and, in the case of CCL8, quantitative, inclusion in the manuscript. However, I am not convinced by the reported fits and the respective Kd values for the other chemokines. For example, CCL5 binding is increased around

three-fold when increasing the concentration from 100 to 500nM, which is not consistent with the fitted K_D s of 54 and 106nM. This can be compared to CCL8 where binding is increased by around 50% when CCL8 concentration is increased from 100 to 500nM, which indicates a higher affinity for CCL8 than CCL5. However, according to the fits CCL5 has higher affinity, which does not pass the eye test.

Multiple factors could affect this. For example, oligomerization of CCL5 will be increased at the higher concentration and dimer or higher oligomer binding, which has been previously observed for CCL5, would give a larger signal response. In general, the curves at high chemokine concentration appears to have slower k_{obs} with signal accumulating over time than those at lower chemokine concentration, which is a sign that there are oligomer or other potentially non-specific contributions at high chemokine concentrations. For an interaction following pseudo-first order kinetics k_{obs} would be expected to increase with increasing chemokine concentration, as seen for CCL8.

In summary, the BLI data convincingly shows that the chemokines bind the peptide but (except for CCL8) the use of a 2:1 binding model in the analysis and the reporting of multiple exact K_D values is, in my opinion, not appropriate based on the presented data.

Response: We thank the reviewer for the critical analysis of our BLI data. We agree that chemokine oligomerization is the most parsimonious explanation for the observed binding profiles which prevent 1:1 fitting of the data. In the revised version of the manuscript, we present all dose-response curves obtained in the experiment instead of excluding them if they did not fit the binding model (revised Fig. 4) and also show the 1:1 fit as dotted lines. We only report K_D and its standard error if the 1:1 fits were of good quality, and have removed the 2:1 fitting analysis. We have re-written the BLI results section (lines 165-186) accordingly.

1. Yin, R., Feng, B.Y., Varshney, A. & Pierce, B.G. Benchmarking AlphaFold for protein complex modeling reveals accuracy determinants. *Protein Sci* **31**, e4379 (2022).
2. Denisov, S.S. et al. Structural characterization of anti-CCL5 activity of the tick salivary protein evasin-4. *The Journal of biological chemistry* **295**, 14367-14378 (2020).